# Doughnut of social and planetary boundaries monitors a world out of balance

Andrew L. Fanning[1,2,3] & Kate Raworth[1,4,5]

The doughnut-shaped framework of social and planetary boundaries (the 'Doughnut') provides a concise visual assessment of progress towards the goal of meeting the needs of all people within the means of the living planet[1–3]. Here we present a renewed Doughnut framework with a revised set of 35 indicators that monitor trends in social deprivation and ecological overshoot over the 2000–2022 period. Although global gross domestic product (GDP) has more than doubled, our median results show a modest achievement in reducing human deprivation that would have to accelerate fivefold to meet the needs of all people by 2030. Meanwhile, the increase in ecological overshoot would have to stop immediately and accelerate nearly two times faster towards planetary boundaries to safeguard Earth-system stability by 2050. Disaggregating these global findings shows that the richest 20% of nations, with 15% of the global population, contribute more than 40% of annual ecological overshoot, whereas the poorest 40% of countries, with 42% of the global population, experience more than 60% of the social shortfall. These trends and inequalities reaffirm the case for overcoming the dependence of nations on perpetual GDP growth[4,5] and reorienting towards regenerative and distributive economic activity—within and between nations—that assigns priority to human needs and planetary integrity.

In the twentieth century, the predominant conception of progress came to focus on raising standards of living through the pursuit of economic growth, measured as an increase in the GDP of nations[6,7]. An emerging twenty-first century conception of progress is focused on far more holistic ambitions, such as well-being[8], sustainable development[9], prosperity[10] and planetary health[11]. These emergent Western-based conceptions reflect notions of socio-ecological balance that have long been central to many Indigenous worldviews[12]. Integral to all of them is a commitment to meeting the essential needs of all people while protecting the stability of Earth's life-supporting systems, on which all life depends.

The doughnut-shaped framework of social and planetary boundaries (commonly known as the Doughnut) provides a concise visual illustration and quantification of progress towards achieving these dual objectives[1–3]. In this Analysis, we present a strengthened framework of boundaries and indicators for the Doughnut at the global scale, bringing together recent advances in planetary boundaries science[13] and in the availability of data monitoring social outcomes worldwide[14]. Also, we explore inequalities that tend to be masked by such global aggregates by comparing the performance of countries clustered by income level, using a comparable set of social indicators coupled with recent estimates of 'downscaled' planetary boundaries[15].

At its conceptual core, the Doughnut framework consists of the space between two concentric rings: the inner ring represents a social foundation, below which lies critical human deprivation, and the outer ring represents an ecological ceiling, beyond which lies critical planetary degradation. Between these two rings lies a doughnut-shaped area that delimits humanity's twenty-first century ambition to safeguard the stability of our planetary home while ensuring that no one falls short of meeting their essential needs—together creating the minimal conditions for defining an ecologically safe and socially just space for humanity (Extended Data Fig. 1).

First created in 2012 (ref. 1), this framework was updated in 2017 (refs. 2,3) to reflect evolving understanding and measurement of both critical human deprivation and planetary degradation. These initial analyses suggested that humanity was not living within the Doughnut at the global scale. Over the same period, the Doughnut framework has attracted considerable interest as a goal for humanity across many facets of society, such as among educators[16,17], community organizers[18,19], social enterprises[20,21] and governments at subnational[22], national[23,24] and international levels[25].

Here we propose a third iteration of the global Doughnut, with a revised set of 21 dimensions (Extended Data Fig. 2), measured by 35 indicators of social shortfall and ecological overshoot. We also present our efforts to transform the Doughnut from being a static snapshot in a single year to an annual monitor of global social and ecological health in the twenty-first century, by incorporating time-series data from 2000 to 2022 for as many indicators as possible ($N = 30$).

At the global scale, we measure social shortfall and ecological overshoot by comparing social indicators with respect to minimum social standards in the Doughnut's social foundation and ecological indicators with respect to planetary boundaries in the Doughnut's ecological

[1]Doughnut Economics Action Lab, Oxford, UK. [2]School of Earth and Environment, University of Leeds, Leeds, UK. [3]Instituto Universitario de Investigación Marina, University of Cádiz, Puerto Real, Spain. [4]Environmental Change Institute, Oxford University Centre for the Environment, Oxford, UK. [5]Centre for Economic Transformation, Amsterdam University of Applied Sciences, Amsterdam, The Netherlands. ✉e-mail: andrew@doughnuteconomics.org

ceiling. It is beyond our scope here to provide a critical assessment of the framework of nine planetary boundaries, although we note that it has been the subject of considerable scientific scrutiny[26] and its limitations may affect our assessment (see Methods for further details). The Doughnut's social foundation has also been critically assessed by some authors[27–29], although the literature in this area has only recently begun to be reviewed systematically[30].

It is essential to note that the global population of 'humanity' depicted in the Doughnut is not a single entity. Previous research has shown that no country is meeting the needs of all its residents with a level of resource use that could be sustainably extended to all people[31,32], but at the same time, nations vary enormously in terms of their global responsibility for overshooting planetary boundaries, which is being driven disproportionately by extractive production and overconsumption by the affluent[15,33,34]. Furthermore, nations have been shown to be highly diverse in their capacity to achieve the social foundation for all their residents, especially in the Global South, in which the legacy of colonialism looms large[32,35,36].

Towards making such disparities visible, we differentiate social shortfall and ecological overshoot in 2017 for 27 comparable indicators across three broad clusters of countries, grouped by income per capita percentiles (the poorest 40% of countries, middle 40% of countries and richest 20% of countries). This approach builds on methods to downscale the global Doughnut using consumption-based 'footprint' indicators that account for the outsourcing of upstream environmental burdens enabled by international trade[31,32].

Our Analysis integrates earlier work by combining a global perspective[2,3] with time-series and cross-country comparative approaches[15,32]. At the global scale, we provide further insights by estimating the historical trends observed across each social and ecological indicator and illustrating how these relate to the ambition needed for humanity to live within the Doughnut by 2050. At the country-cluster scale, our Analysis provides a new integrated comparison of the shares of global social shortfall held by each country cluster alongside their respective contributions to global ecological overshoot. We also ensure that downscaled social indicators track population shares in shortfall using the same dimensions and deprivation-based indicators as the global Doughnut, and we compare country-cluster environmental footprints with respect to four downscaled per capita boundaries in a coherent visual framework that is consistent with the broader planetary boundaries framework. The data collection and analysis procedures are described in Methods, with indicator-specific details provided in the Supplementary Information, including an overview of all data sources in Supplementary Table 1 (for social indicators) and Supplementary Table 2 (for ecological indicators).

## Current global status

If human progress this century depends on eliminating social shortfall and ecological overshoot simultaneously[2], our latest synthesis underscores that the world is still far from securing it. Billions of people are falling short of meeting their most essential needs, whereas humanity's ecological imprint on the living planet is now overshooting at least six of the nine planetary boundaries (Fig. 1).

The share of the global population experiencing social shortfall varies widely across the 12 dimensions of the Doughnut's social foundation and their respective indicators with available data ($N = 21$). The status of these social indicators in 2022 ranges from 9% of people lacking access to electricity to 75% of people stating they perceive widespread corruption, with a median level of social shortfall comprising 35% of the global population—equivalent to around 3 billion people. In general, there are lower levels of shortfall for social dimensions that measure access to physical necessities, such as energy and food, and higher levels of shortfall for dimensions that measure the strength of social fabric, such as social cohesion and political voice.

Meanwhile, the variation in overshoot beyond the Doughnut's ecological ceiling is wider than that of the social foundation, across the nine planetary boundaries and their respective indicators ($N = 13$). The status of these ecological indicators in 2022 ranges from around 50% below the stratospheric ozone boundary to more than ten times beyond the safe boundaries for chemical pollution and species extinction, with a median level of overshoot that is nearly two times beyond the ecological ceiling (96%). The transgressions of critical Earth-system processes revealed by planetary boundaries science are concerning, given that these processes underpin the stability of the Holocene-like conditions on which all life fundamentally depends.

## Twenty-first century trends

Over the first two decades of the twenty-first century, there has been a concerning divergence between social and ecological trends at the global scale (Fig. 2). Humanity has generally made progress towards achieving the Doughnut's social foundation, although this improvement has been modest, given the extent of deprivation remaining. At the same time, humanity's collective pressure on the planet has substantially worsened since the early 2000s and now transgresses at least six of the nine planetary boundaries.

The number of social indicators with most of humanity in shortfall has been halved since the early 2000s (from ten indicators to five indicators) and 13 of the 21 indicators show discernible reductions in shortfall. The five largest social improvements are for internet connectivity, health services coverage, child survival, safe sanitation and clean indoor fuels, with 24–56% of humanity escaping shortfall in these areas over the past two decades, depending on the indicator. However, there are five indicators that show little change over the period (undernourishment, youth unemployment, lack of safe drinking water, lack of social support and perceptions of corruption), whereas food insecurity and political voice have worsened.

The median share of humanity experiencing social shortfall declines from 47% in 2000 to 35% in 2022 and the interquartile range across the social indicators narrows substantially from 27–73% of the population to 22–42% over the same period (Extended Data Fig. 3). Despite these generally positive trends, at least 2 billion people still fall below the Doughnut's social foundation across most of the indicators (15 out of 21 indicators show at least 25% of the global population in shortfall in 2022).

Meanwhile, the ten ecological indicators with available time-series data show a considerable worsening of conditions with respect to the Doughnut's ecological ceiling since the early 2000s, with the notable exception of ozone-layer depletion, which is roughly stable over the period. Two indicators increase their extent of overshoot by 70–80% between 2000 and 2022 (nitrogen use and green-water disruption), whereas four indicators increase even further, more than doubling their respective levels of overshoot over the same period ($CO_2$ concentration, radiative forcing, hazardous chemicals and phosphorus). There are two indicators that increase their extent of overshoot by 10–25% compared with 2000 levels (loss of forest cover and human appropriation of net primary productivity (HANPP)), moving further away from safe Holocene-like conditions more slowly than the others, but still doing so rapidly on the geological timescale of the Earth system.

The median level of overshoot beyond the ecological ceiling increases from 75% in 2000 to 96% in 2022 and the interquartile range across the indicators widens substantially from 27–119% beyond the ecological ceiling to 61–213% over the same period (Extended Data Fig. 4). The ocean acidification indicator is still below its boundary in 2022 but more than three-quarters of the gap below the boundary in 2000 was closed over the 23-year period, leaving just 6% of its safe space remaining with respect to the Holocene baseline, and closing quickly. Despite these highly concerning trends, they may yet underestimate the overall increase in degradation because we could not assess changes over time for three ecological indicators owing to a

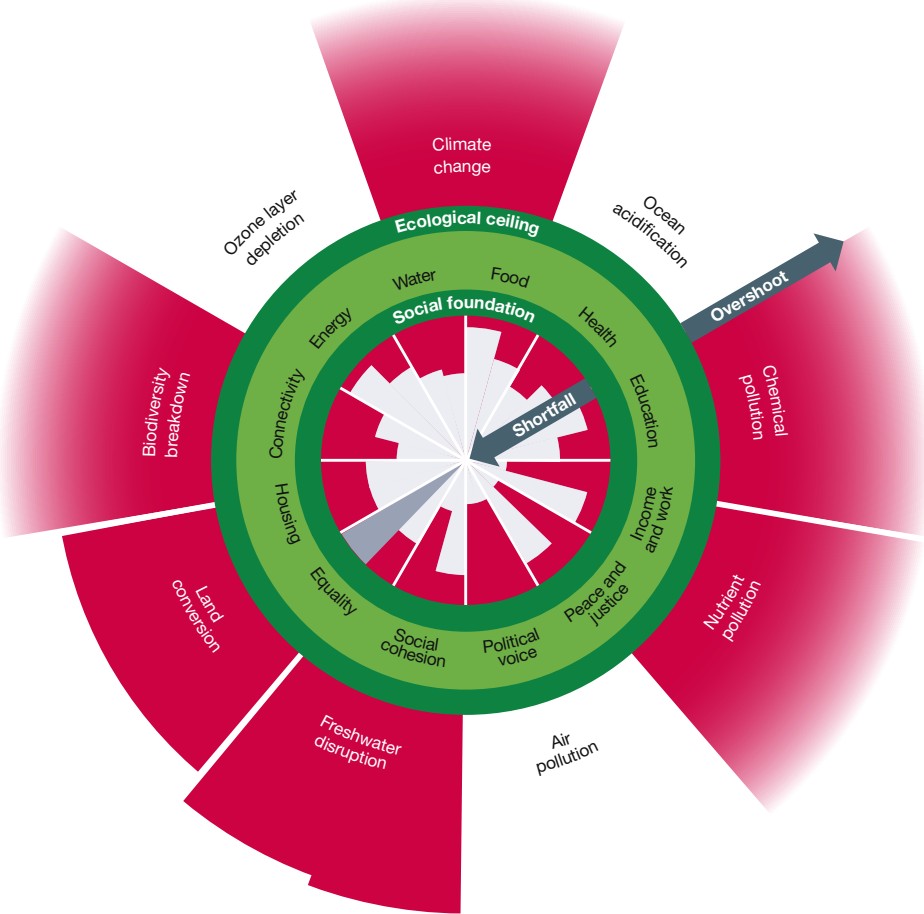

**Fig. 1 | Current global status of shortfall and overshoot in the Doughnut of social and planetary boundaries.** Values shown are for 2022. Social wedges (inner ring) show the status of humanity relative to minimum social standards and ecological wedges (outer ring) show Earth-system status relative to planetary boundaries. Red wedges show a shortfall below the social foundation or an overshoot beyond the ecological ceiling. The grey wedge indicates missing data. The centre of the plot represents total human deprivation (for the social indicators) and the pre-industrial Holocene baseline (for the ecological indicators). Values are proportional to the length of each wedge, which leads to a quadratic scaling of wedge area that may lead some readers to perceive small changes as more significant than they are. To address this limitation, we provide Fig. 2 with a bar-chart representation that avoids this quadratic scaling issue. Wedges with a faded edge extend beyond the chart area. A maximum of two indicators are included in each social or ecological dimension. Ideally, there would be no red wedges below the social foundation or above the ecological ceiling. See Tables 1 and 2 for details on all of the social and ecological indicators, including their respective boundaries. Adapted from ref. 2, CC BY 4.0.

lack of data (disruption to blue-water flows, species extinction rate and interhemispheric aerosols).

## Progress far off-course

Tables 1 and 2 present the Doughnut's revised social and ecological indicators, including historical levels and trends since the early 2000s for each indicator, alongside scenario trends that would be sufficient to eliminate social shortfall by 2030 and ecological overshoot by 2050. We estimated annual historical time trends for each indicator statistically using ordinary least squares regression. Scenario trends were calculated linearly as the annual percentage change required between 2022 values and zero for each indicator by 2030 (for social indicators) and 2050 (for ecological indicators). The procedures are detailed in Methods and indicator-specific statistical estimates are provided in the Supplementary Data.

On the basis of the historical trends observed over the first two decades of the twenty-first century, we find that social shortfall improves by 0–1 percentage points (%pt) per year, based on the interquartile range estimated over the 2000–2022 period for each indicator (median: 0.5%pt per year). Meanwhile, the interquartile range across the ecological indicators shows overshoot worsening by 1–5%pt per year over the same period (median: 2.8%pt per year).

There are statistically significant improving trends in reducing social shortfall for 17 social indicators (*P* < 0.01), ranging from less than 0.2%pt per year (youth unemployment and income inequality) to nearly 3%pt per year (health service coverage and internet connectivity; Table 1 and Extended Data Fig. 5). Although these social indicators show improving performance, none of their historical trends are sufficiently rapid to eliminate social shortfall by 2030, in line with the ambition of the United Nations (UN) Sustainable Development Goals (SDGs). To achieve such an ambition would require rates of improvement across these social indicators to increase by a median of 4.9 times (interquartile range: 3.6–14.8 times) in comparison with 2000–2022 trends.

The three best-performing social indicators would need to accelerate historical rates of improvement by 25–80% to eliminate shortfalls by 2030 (health service coverage, electricity access and internet connectivity). Meanwhile, six laggard indicators would need to accelerate historical rates of improvement by at least tenfold to reach the Doughnut's social foundation by 2030 (societal poverty, youth unemployment, unsafe drinking water, informal housing, income inequality and perceptions of corruption). The remaining eight indicators lie in between

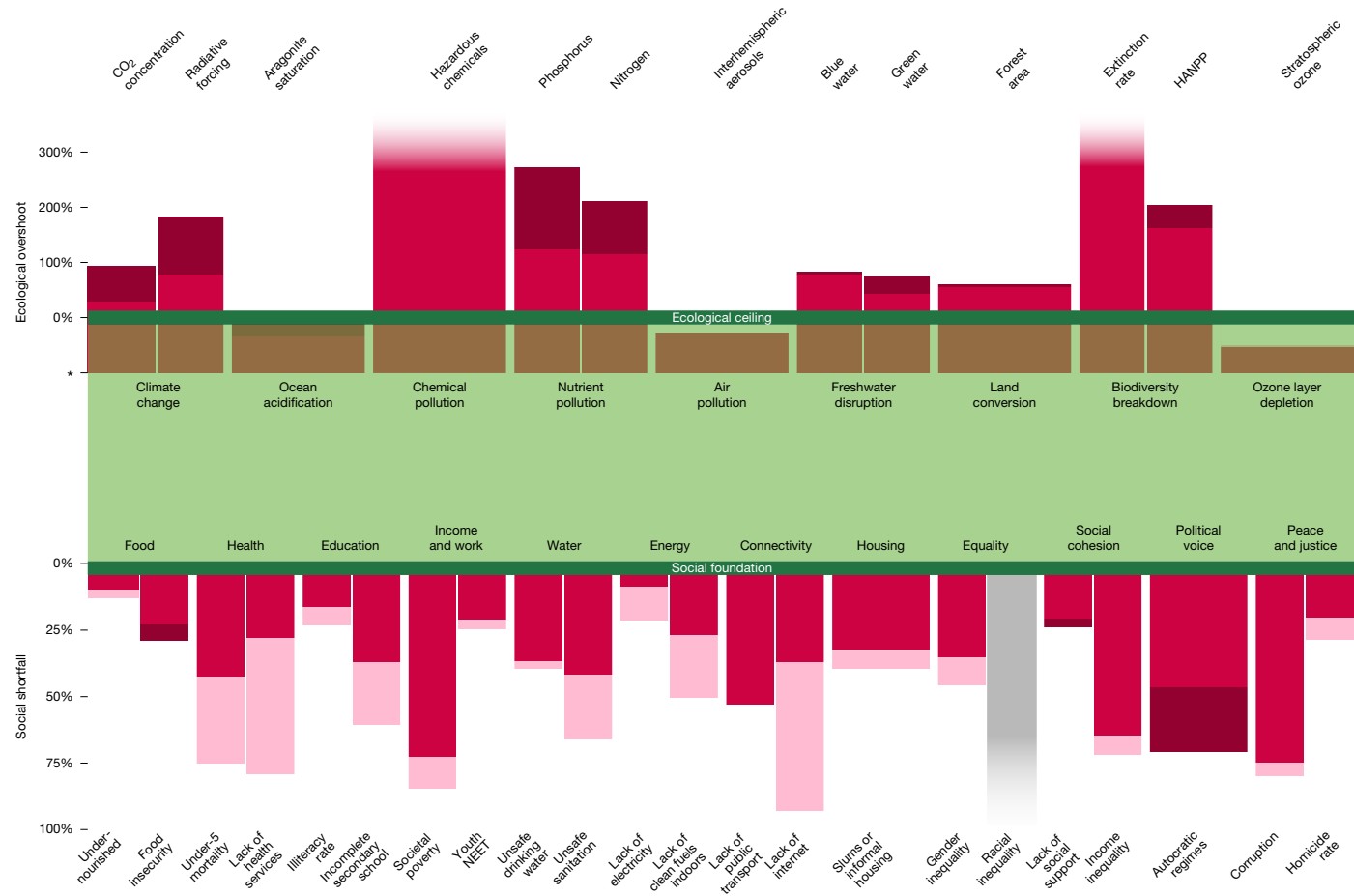

**Fig. 2 | Global change in social shortfall and ecological overshoot between 2000–2001 and 2021–2022.** Social shortfall is measured with respect to the social foundation, for which zero indicates no shortfall. Ecological overshoot is measured with respect to planetary boundaries, for which zero indicates no overshoot and * indicates the pre-industrial Holocene baseline. Values are shown as 2-year averages at the start and end of the analysis period except for stratospheric ozone values, which are shown as 5-year averages owing to high annual variability. Light red bars show the reduction in shortfall or overshoot between the start and end periods. Dark red bars show the increase in shortfall or overshoot between the start and end periods. The grey bar shows missing data. Overshoot bars with a faded edge extend beyond the chart area. No change in shortfall or overshoot is shown for four indicators owing to a lack of time-series data (lack of public transport, blue-water disruption, extinction rate and air pollution). Owing to a lack of earlier data, three indicators start in 2005 (youth not in employment, education or training (NEET), lack of social support and perceptions of corruption), whereas data on food insecurity starts in 2015. See Tables 1 and 2 for more details on the social and ecological indicators, including their boundaries.

these extremes and would need to accelerate rates of improvement 3–9 times faster than 2000–2022 trends to eliminate social shortfall by 2030.

Notably, two social indicators show significantly worsening trends (food insecurity and autocratic regimes) that would need to stop immediately and rapidly accelerate in reverse to eliminate shortfalls by 2030 (at 3.3%pt per year and 8.8%pt per year, respectively). Although the social support indicator also shows a slightly worsening trend, this finding is not statistically significant owing to a lack of directionality over the 2000–2022 period.

From an ecological perspective, there are highly significant worsening trends in ecological overshoot across nine out of ten indicators with available time-series data ($P < 0.001$), ranging from 0.3%pt per year (forest conversion) to more than 80%pt per year (hazardous chemicals; Table 2 and Extended Data Fig. 6). The median (and interquartile range) rate of increasing overshoot across these indicators is 3.9 (2.4–5.9)%pt per year. A scenario to eliminate ecological overshoot by 2050 across these indicators would mean immediately stopping their rapid rates of degradation and accelerating in reverse at a median rate of regeneration that is nearly two times faster, or 6.9 (3.2–8.1)%pt per year for the next three decades (except for ocean acidification, which could remain stopped below its boundary). We note that the

likelihood of following such a regenerative pathway seems to be very low, on the basis of historical trends, and may not even be physically possible by 2050 for some ecological indicators owing to lags in the Earth-system, such as for atmospheric concentration of carbon dioxide and for species-extinction rates[37].

## Inequalities in shortfall and overshoot

We investigate inequalities in our global results by differentiating three broad clusters of countries with available data—the poorest 40% of countries, middle 40% of countries and richest 20% of countries—based on average national income per capita across 193 countries over the 2000–2022 period (Extended Data Fig. 7). The social indicators can be disaggregated into the three country clusters in a manner that is directly comparable with the global results ($N = 21$). The ecological indicators, however, are related to Earth-system processes that cannot be disaggregated to a national scale in a directly comparable manner[26]. Scholars have developed methods that translate planetary boundaries into global proxies, which can then be divided up among individual countries based on a sharing principle, such as equality or sovereignty[38]. We collected national consumption-based 'environmental footprint' data across 168 countries in the year 2017 for six indicators (carbon

**Table 1 | The social foundation and its indicators of shortfall**

| Dimension | Indicator | Shortfall (%) | | Historical trend | To eliminate shortfall by 2030 |
|---|---|---|---|---|---|
| | (% of global population, unless otherwise stated) | 2000–2001 | 2021–2022 | (%pt per year) | (%pt per year) |
| Food | Population undernourished | 13 | 10 | −0.2** (improving) | −1.2 |
| | Population with moderate to severe food insecurity | (23) | 29 | +1.1** (worsening) | −3.6 |
| Health | Population living in countries with under-5 mortality rate exceeding 25 per 1,000 live births | 75 | 42 | −1.4** (improving) | −5.3 |
| | Population living in countries without high coverage of essential health services (Universal Health Coverage Index score less than 60 out of 100) | 79 | 28 | −2.8** (improving) | −3.5 |
| Education | Adult population (aged 15+ years) who are illiterate | 23 | 16 | −0.4** (improving) | −2.0 |
| | Young adult population (aged 21–23 years) with incomplete upper secondary education | 61 | 37 | −1.3** (improving) | −4.6 |
| Income and work | Population living below the societal poverty line, set at half their country's median household income or at least $15 a day | 85 | 73 | −0.6** (improving) | −9.1 |
| | Population of young people (aged 15–24 years) not in employment, education or training | (25) | 21 | −0.1* (improving) | −2.6 |
| Water and sanitation | Population lacking access to safely managed drinking water | 39 | 37 | −0.1** (improving) | −4.6 |
| | Population lacking access to safely managed sanitation | 66 | 42 | −1.2** (improving) | −5.2 |
| Energy | Population lacking access to electricity | 21 | 8 | −0.6** (improving) | −1.1 |
| | Population lacking access to clean fuels and technologies for cooking, heating and lighting | 50 | 27 | −1.2** (improving) | −3.3 |
| Connectivity | Urban population lacking convenient access to public transport | – | 53 | – (not known) | −6.6 |
| | Population not accessing the internet | 93 | 37 | −2.6** (improving) | −4.6 |
| Housing | Urban population living in slums or informal settlements | 40 | 32 | −0.4** (improving) | −4.0 |
| Equality | Population-weighted score on the Gender Inequality Index (global gap between women and men in terms of reproductive health, empowerment and employment) | 46 | 35 | −0.5** (improving) | −4.4 |
| | Racial inequality (no global indicator tracks racial and ethnic equality gaps in social outcomes at present) | – | – | – (not known) | – |
| Social cohesion | Population stating that they are without someone to count on in times of trouble | (21) | 24 | +0.1 (no change) | −3.0 |
| | Population living in countries with a Palma ratio of 2 or more (the income share of the richest 10% of people relative to the poorest 40%) | 72 | 65 | −0.2** (improving) | −8.1 |
| Political voice | Population living in countries governed by an autocratic regime | 46 | 71 | +1.0** (worsening) | −8.8 |
| Peace and justice | Population stating that they perceive widespread corruption in government and business | (80) | 75 | −0.5** (improving) | −9.3 |
| | Population living in countries with a homicide rate of 5 or more per 100,000 | 29 | 20 | −0.5** (improving) | −2.5 |

Values in the '2000–2001' and '2021–2022' columns are reported as two-year averages. Percentages in parentheses are available from a later start year for food insecurity (2015); youth not in employment, education or training, social support and corruption (2005). Historical trends are estimated using ordinary least squares regression and two-sided hypothesis tests with statistical significance indicated at the 99% (*) and 99.9% (**) levels. Pathways to eliminate shortfall by 2030 are calculated as percentage changes between 2021–2022 levels of shortfall and zero. See Supplementary Discussion 1 for indicator details and Supplementary Table 1 for sources.

dioxide, nitrogen, phosphorus, blue water, species loss and HANPP)[15] and calculated population-weighted averages for the three country clusters. The per capita footprint indicators for each country cluster were compared with equality-based per capita budgets collected from the same source[15], which are related to four planetary boundaries (climate change, nutrient pollution, freshwater disruption and biodiversity breakdown; see Methods).

We find that social shortfall improves and ecological overshoot worsens as income levels increase across the three country clusters (Fig. 3), which is broadly consistent with previous cross-country comparisons[4,31,32]. The richest 20% country cluster has the least social shortfall while having the greatest ecological impact across all six indicators, with environmental footprints that are 1.3–12.4 times larger than those of the poorest 40% of countries. By contrast, the poorest 40% country cluster has the least ecological overshoot while having the greatest social shortfall across 19 out of 21 indicators, with a share of population in deprivation that is 1.3–94.0 times larger than that of the richest 20% of countries.

## Table 2 | The ecological ceiling and its indicators of overshoot

| Dimension | Indicator (and planetary boundary) | Value (and % overshoot beyond boundary) | | Historical trend | To eliminate overshoot by 2050 |
|---|---|---|---|---|---|
| | | 2000–2001 | 2021–2022 | (%pt per year) | (%pt per year) |
| Climate change | Atmospheric carbon dioxide concentration, parts per million (at most 350 ppm $CO_2$) | 370 ppm (28%) | 416 ppm (94%) | +3.1** (worsening) | −3.4 |
| | Human-induced radiative forcing at the top of the atmosphere, Watt per square metre (at most 1 W m$^{-2}$) | 1.8 W m$^{-2}$ (78%) | 2.8 W m$^{-2}$ (183%) | +5.5** (worsening) | −6.5 |
| Ocean acidification | Average saturation state of aragonite at the ocean surface (at least 80% of pre-industrial saturation state of 3.44 $\Omega_{arag}$) | 2.99 $\Omega_{arag}$ (−34%) | 2.80 $\Omega_{arag}$ (−6%) | +1.3** (worsening) | (within boundary) |
| Chemical pollution | Production of hazardous chemicals, millions of tonnes per year (at most 5% of the 1,200 Mt of total chemicals produced in year 2000) | 933 Mt (1,455%) | 1,964 Mt (3,174%) | +81.8** (worsening) | −113 |
| Nutrient pollution | Phosphorus applied to land as fertilizer, millions of tonnes per year (at most 6.2 Mt per year) | 14 Mt (123%) | 23 Mt (273%) | +7.1** (worsening) | −9.7 |
| | Nitrogen applied to land as fertilizer, millions of tonnes per year (at most 62 Mt per year) | 134 Mt (116%) | 193 Mt (212%) | +4.6** (worsening) | −7.6 |
| Air pollution | Asymmetry between Earth's hemispheres of sunlight reaching the surface, owing to differences in atmospheric particle concentration (at most 0.1 inter-hemispheric difference in Aerosol Optical Depth) | 0.08 AOD (−29%) | 0.08 AOD (−29%) | – (not known) | (within boundary) |
| Freshwater disruption | Proportion of land area with human-induced disturbance of blue-water flow deviating from Holocene variability (at most 10.2%) | 18.2% dev. (78%) | 18.2% dev. (78%) | – (not known) | −2.7 |
| | Proportion of land area with root-zone soil moisture deviating from Holocene variability (at most 11.1%) | 15.9% dev. (43%) | 19.3% dev. (74%) | +2.5** (worsening) | −2.6 |
| Land conversion | Area of forested land as a proportion of forest-covered land before human alteration (at least 75% of 64 million square kilometres) | 39 Mkm$^2$ (55%) | 38 Mkm$^2$ (61%) | +0.3** (worsening) | −2.2 |
| Biodiversity breakdown | Rate of species extinctions per million species years (at most 10 E/MSY) | 100 E/MSY (900%) | 100 E/MSY (900%) | – (not known) | −32 |
| | Human appropriation of net primary productivity, billions of tonnes of carbon per year (at most 10% of 55.9 GtC) | 15 GtC (162%) | 17 GtC (204%) | +2.0** (worsening) | −7.3 |
| Ozone-layer depletion | Concentration of ozone in the stratosphere, Dobson units (at most 5% decrease with respect to 1964–1980 value of 290 DU) | 283.0 DU (−50%) | 283.4 DU (−53%) | +0.1 (no change) | (within boundary) |

Values in the '2000–2001' and '2021–2022' columns are reported as two-year averages except for stratospheric ozone concentration values, which are reported as five-year averages owing to high annual variability (2000–2004 and 2018–2022). Historical trends are estimated using ordinary least squares regression and two-sided hypothesis tests with statistical significance indicated at the 99% (*) and 99.9% (**) levels. Pathways to eliminate overshoot by 2050 are calculated as the percentage change between 2020–2021 levels of overshoot and zero. See Supplementary Discussion 2 for indicator details and Supplementary Table 2 for sources.

The extent of ecological overshoot in the richest 20% of countries exceeds the other two country clusters by a large margin, especially for carbon dioxide and species-loss footprints (Fig. 4a). The richest 20% and the middle 40% country clusters both overshoot five of the six per capita boundaries, but the median extent of overshoot is nearly three times larger in the richest 20% of countries (273% and 96% overshoot, respectively) (Extended Data Fig. 9). Meanwhile, the poorest 40% country cluster is within all per capita boundaries except for nitrogen and phosphorus (median: 1% overshoot). None of the country clusters transgress the per capita boundary for freshwater disruption, although we note that the proxy indicator of global blue-water consumption per capita does not reflect regional water scarcities[15].

The extent of social shortfall in the poorest 40% of countries tends to be larger than the other clusters, especially for dimensions more closely related to physical necessities, such as health, energy and housing (Fig. 4b). The median share of population in shortfall in this low-income cluster is double that of the middle 40% of countries and nine times larger than that of the richest 20% of countries (60%, 29% and 6.6% shortfall, respectively) (Extended Data Fig. 8). The two peace and justice indicators are exceptions to this general pattern, with

population reporting widespread perceptions of corruption highest in the middle-40% of countries (82%) and the population living in countries with a homicide rate of 5 or more (per 100,000 people) is highest in the richest 20% of countries (32%). Notably, the latter finding is driven by just two countries (out of 35) that exceed the homicide rate boundary in this wealthy group—the Bahamas and the United States—with the relatively large population of the United States playing a dominant role.

Our disaggregated analysis reveals clear inequalities across the three country clusters in terms of their respective contributions to global levels of ecological overshoot and social shortfall in 2017 (Extended Data Fig. 10). The richest 20% country cluster includes only 15% of the world's population but it contributes a disproportionately large 26–73% share to global ecological overshoot on an annual basis, depending on the indicator (median: 44%). Simultaneously, this wealthy group holds a relatively small 0–21% share of global social shortfall (median: 2%). By contrast, the poorest 40% country cluster contributes a negligible 0–18% share to global ecological overshoot (median: 4%), which is far below its 42% share of world population. However, this poorer group holds a disproportionately large 27–97% share of global social shortfall

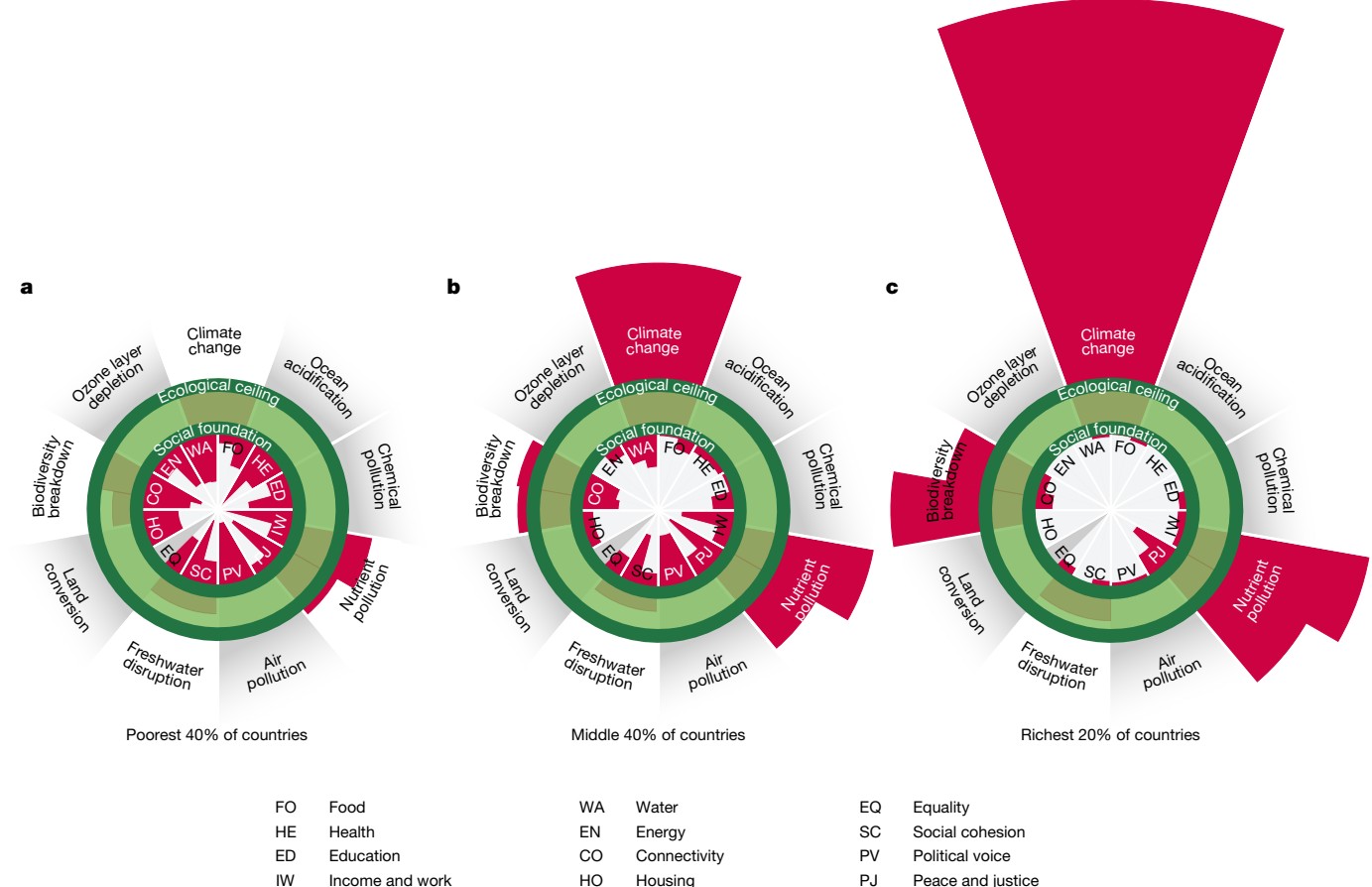

a — Poorest 40% of countries

b — Middle 40% of countries

c — Richest 20% of countries

| | | | | | |
|---|---|---|---|---|---|
| FO | Food | WA | Water | EQ | Equality |
| HE | Health | EN | Energy | SC | Social cohesion |
| ED | Education | CO | Connectivity | PV | Political voice |
| IW | Income and work | HO | Housing | PJ | Peace and justice |

**Fig. 3 | Status of shortfall and overshoot in a disaggregated Doughnut framework for three country clusters. a**, Poorest 40% of countries. **b**, Middle 40% of countries. **c**, Richest 20% of countries. Values shown for each indicator are population-based aggregates of national values in 2017 (except for the public transport indicator, which shows equivalent values in 2020 owing to a lack of earlier data), with countries clustered by percentile level of income per capita. Social wedges (inner ring) show the status of each country cluster's population relative to minimum social standards and ecological wedges (outer ring) show the status of each country cluster's consumption-based environmental 'footprints' relative to downscaled per capita boundaries. The colours used are as per Fig. 1. Social wedges start at the inner edge of the social foundation (which represents zero human deprivation), whereas ecological wedges start at the outer edge of the social foundation (which represents zero environmental footprint). Values are proportional to the length of each wedge, which leads to a quadratic scaling of wedge area that may lead some readers to perceive small changes as more significant than they are. To address this limitation, we provide Fig. 4 with a bar-chart representation that avoids this quadratic scaling issue. A maximum of two indicators are included to illustrate country-cluster performance in each social or ecological dimension. See Supplementary Table 1 for details on the social indicators, Supplementary Table 2 for details on the environmental footprint indicators, including their respective downscaled per capita boundaries, and Supplementary Data for data for all country clusters.

(median: 63%). Meanwhile, the middle 40% country cluster contributes a 27–58% share to global ecological overshoot (median: 52%) and holds a 29–53% share of global social shortfall (median: 33%), which are both more in proportion to its 43% share of world population.

## Redefining and reorienting progress

The Doughnut framework informs three critical transformations that we believe are essential to shift to a twenty-first century pattern of progress: its conception, its metrics and its directionality. Last century's dominant conception of progress, based in material living standards, is being replaced with a far more holistic approach, for which the Doughnut's concentric circles provide a concise visual representation: meeting the essential needs of all people within the means of the living planet[4]. The metrics of progress are likewise changing, beyond a predominant focus on the monetary metric of GDP, to consider an array of social and ecological dimensions that aim to monitor human well-being and planetary health in their own terms[5,39]; the Doughnut combines these into a coherent, visual dashboard of 35 indicators that can be continually updated and improved as better metrics become available. The directionality of progress is also shifting, away from

perpetual growth towards a dynamic that will define success this century: eliminating global social shortfall and ecological overshoot simultaneously. By transforming the Doughnut into an annual monitor of humanity's trajectory and tracking it over the first two decades of the twenty-first century, we make visible the current directionality of this dynamic in relation to what is required.

Although global GDP doubled between 2000 and 2022, the Doughnut's trend analysis underscores a very different story: only modest improvements were achieved in reducing social shortfalls worldwide, whereas ecological overshoot increased rapidly, disrupting the critical planetary processes on which all life depends. Economic policy-making that assigns priority to perpetual economic growth has been failing to bring humanity into the Doughnut's safe and just space. This reaffirms calls from post-growth scholars—ranging from degrowth[40] to well-being economy[8]—for a deep renewal of both economic theory and practice, including by overcoming nations' structural dependency on GDP growth, so that they can instead reorient towards ecologically regenerative and socially distributive economic policies and outcomes[3–5]. Such reorientation must acknowledge country-specific differences in priorities and responsibilities[32,41], given the disproportionate contribution to global ecological overshoot by the richest,

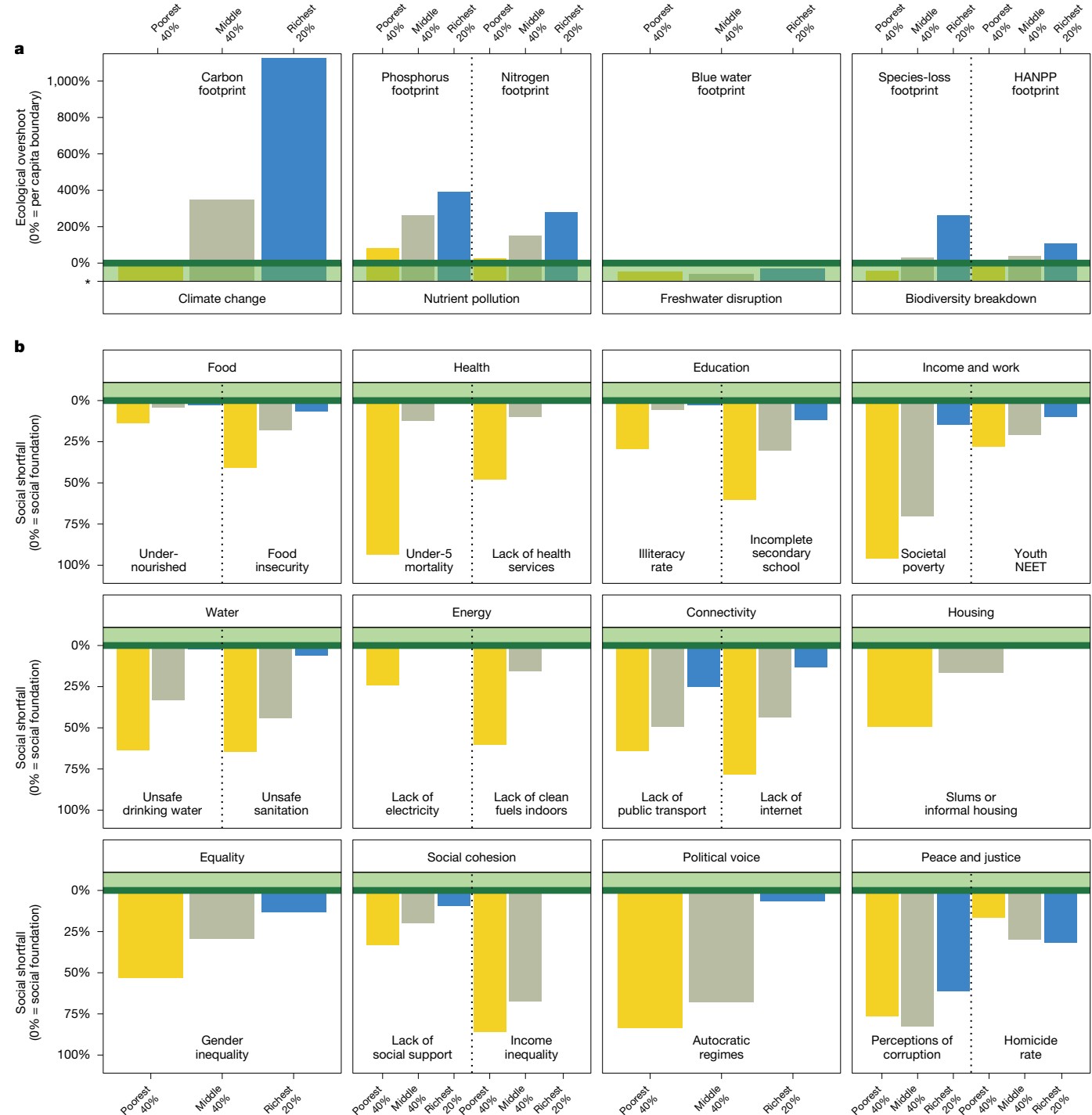

**Fig. 4 | Levels of social shortfall and ecological overshoot by country cluster.** **a**, Ecological dimensions and footprint indicators relative to downscaled per capita boundaries ($N$ = 6). **b**, Social dimensions and indicators relative to the social foundation ($N$ = 21). Values shown are for 2017 (except for the public transport indicator, which shows 2020 values owing to a lack of earlier data). Bars are colour-coded by country cluster: poorest 40% of countries (yellow), middle 40% of countries (beige) and top 20% of countries (blue). Dashed vertical lines separate indicators in each social or ecological dimension, if more than one is included. See Supplementary Tables 1 and 2 for more details on the social and ecological indicators, including their respective boundaries, and Supplementary Data for data for all country clusters.

and the disproportionate share of population in deprivation among the poorest, as shown by the Doughnut's country-cluster analysis.

Our Analysis underscores a need to continue deepening research into post-growth economic futures, especially in high-income countries[4]. The data underpinning our analysis can serve as an input into eco-logical macroeconomic modelling that seeks to assess policy options with the potential to decouple human well-being from both ecological overshoot and economic growth[42]. Complementary research into exist-ing policymaking would also be valuable, including studying the early experience of more than 50 city and district governments worldwide that have, since 2019, started to embed the Doughnut framework in their local strategies, policies and processes[22,43].

We note that there are relatively few monitoring alternatives that combine the holistic ambition to track humanity's social and ecological

performance with respect to explicit targets across a dashboard of indicators, as far as we are aware. Although we have not conducted a comparative analysis of the Doughnut's dimensions and indicators with alternative monitoring frameworks in the present study, the most similar alternative at the global scale is the SDG framework, which earlier studies have compared[30,44]. In general, the Doughnut has a less comprehensive suite of targets than the SDGs (35 versus 169) but a more even balance between social and ecological dimensions and a more holistic and intuitive visual representation.

Other complementary frameworks that make social and ecological targets and/or minimum standards explicitly visible include decent living standards[45,46] and safe and just Earth-system boundaries[47,48]. These frameworks tend to focus on quantifying human deprivation in terms of the units of minimum material requirements that a person needs (for example, per capita use of energy, materials and water) and that can also be related directly to ecological boundaries. The Doughnut is distinct to these in its focus on the share of population falling below minimum social standards, which may be defined in the units of material requirements, such as having access to sufficient food supply and electricity infrastructure, or of non-material requirements, such as having sufficient social support and political voice. An advantage of this approach is that the Doughnut's social foundation can accommodate any aspect deemed necessary to avoid human deprivation on its own terms, whether material or non-material, individual or collective. A disadvantage is that there is no directly traceable link between the social and ecological domains in the Doughnut, although frameworks are emerging that track the global material requirements of achieving minimum standards for all of humanity[49], which we believe are highly complementary and important avenues for future research.

Given the urgency of completing the shift to a new pattern of progress, we intend to continue refining and measuring the Doughnut on an annual basis, so that it may serve as a continually relevant and up-to-date monitor of social and ecological trends. That said, we are aware that such regular updates will not, in and of themselves, transform the dominant growth-based approach. Instead, we see this information flow as an extra tool that scholar-activists and practitioners taking a growth-critical approach can use to push for transformative action in their places and institutions. For example, a strategy that we have been pursuing since 2019, together with colleagues at Doughnut Economics Action Lab, is to convene and make visible a global community that is collectively experimenting in diverse ways to put the concepts of 'Doughnut Economics' into practice in many different contexts by reframing economic narratives, engaging in strategic policy influence, collaborating with like-minded innovators and, crucially, sharing lessons for iterative learning[50]. This is just one action-led example that we are deeply familiar with and we emphasize that many more local, national and international initiatives are working collectively to shift the economic focus away from endless growth and towards ensuring human well-being and planetary health this century.

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

## Methods

This section summarizes how we select and analyse social indicators with respect to minimum social standards in the Doughnut's social foundation and ecological indicators with respect to planetary boundaries and downscaled per capita boundaries in the Doughnut's ecological ceiling. A detailed description of methods and data for each indicator is provided in the Supplementary Information.

### Conceptual framework

There have been two previous iterations of the Doughnut framework: the first version was published in 2012 (ref. 1) and the second in 2017 (refs. 2,3). Since the 2017 version of the Doughnut was published, there have been substantial advances in the global availability of data on social outcomes, especially those indicators monitored in the global indicator framework of the UN SDGs[51]. There have likewise been substantial advances in planetary boundaries science, which have been synthesized at the global scale in the 2023 iteration of the planetary boundaries framework[13] and have also seen widespread uptake across scientific disciplines and society[26]. In this Analysis, we draw on these social and ecological advances in understanding to renew and strengthen the Doughnut framework, through a revision of its dimensions and indicators and the inclusion of available time-series data at the global scale. Building on previous efforts to 'downscale' the Doughnut[31,32], we also develop an approach that disaggregates the global Doughnut in a cross-scale comparable framework to reveal inequalities in social shortfall and ecological overshoot across country clusters. See ref. 52 for a comparison with previous iterations of the Doughnut.

### Global time-series data

We collected time-series data over the 2000–2022 period from publicly available international databases, such as the UN's SDG Indicators Database[14] and scientific sources referenced in the most recent planetary boundaries update[13], among others (see Supplementary Information for all data sources). The first year considered in our analysis is 2000, which is the earliest year available in the SDG Indicators Database[14] and coincides with our ambition to monitor humanity's trajectory in the twenty-first century. For comparability and continuity, we aimed to collect time-series data using the same indicators as Raworth[2,3] in the 2017 version of the Doughnut, unless such data were unavailable or more relevant alternative indicators have since become available.

We collected global estimates directly from international databases for the planetary boundary indicators and calculated population-based estimates from large samples of national values for the social indicators, which were also collected from published and publicly available international databases. For the social indicators, we have included large samples of countries throughout, but we note that there is a varying number of countries with data available per indicator, with a median (and interquartile range) of 151 (124–187) countries included and a 90% (83–99%) share of the world's population (Supplementary Table 1). We acknowledge that indicators measured on the basis of different methods, data sources, system boundaries and country coverage may not be fully comparable, but the holistic nature of the Doughnut makes this unavoidable for the time being, given current data availability.

Furthermore, we note that our use of data published by international data sources does not explore uncertainties within the historical estimates of each social and ecological indicator nor do we fully account for possible shifts within indicator-specific reporting systems. We do, however, document all data sources used in our Analysis in Supplementary Tables 1 and 2, providing explicit references to relevant metadata documentation for each social and ecological indicator issued by its respective data provider. Overall, we include a maximum of two indicators for each dimension and select 22 social indicators and 13 ecological indicators in total (compared with 20 social indicators and ten ecological indicators in the 2017 version).

### Establishing the social foundation

To establish the social foundation, we began with the list of social dimensions and indicators in the 2017 Doughnut and investigated their current relevance in light of advances in the global indicator framework monitoring the SDGs, following six criteria adapted from Raworth[2], which postulate that social indicators should:
• Be globally relevant and serve as effective proxies for broader concerns in their respective dimensions.
• Focus on the worst-off, measuring deprivations as a percentage of the population (as opposed to measuring averages).
• Be measured with sufficiently recent data, with extensive international coverage and with time-series monitoring.
• Have officially recognized thresholds of minimum acceptable standards.
• Be monitored in the global indicator framework of the UN SDGs.
• Include at most two indicators for each dimension of the social foundation.

We considered hundreds of indicators in publicly available international datasets, reached out to relevant experts and used our own expertise to narrow down a draft set of social dimensions and indicators that most closely aligned with these ideal criteria. We conducted an internal review of this draft set of dimensions and indicators with colleagues holding a range of perspectives: some with insight on the broad constituents of human flourishing, others with a focus on specific indicators and data sources (see 'Acknowledgements'). We are grateful for the comments and suggestions shared by colleagues during the internal review process but we make no claims of a representative or exhaustive set of relevant insights nor do we claim that all internal reviewers agree with all of our choices—we take full responsibility for the final indicator selection, as shown in Table 1. We acknowledge that the social indicators that we have selected as best-available proxies for broader concerns in their respective dimensions are open to debate, given that other indicators exist, although in practice we have found the six criteria listed above to be limiting. See Supplementary Discussion 1 for further details on each social indicator and Supplementary Table 1 for data sources.

### Establishing the ecological ceiling

At the global scale, the dimensions of the Doughnut's ecological ceiling are defined by the nine critical Earth-system processes specified in the planetary boundaries framework[13,53,54]. The planetary boundaries framework has seen a very large uptake across academic disciplines, policy circles and the general public[26] but it has also been the subject of considerable scientific debate, which has surfaced many critiques[55] and responses since the original formulation in 2009. Although it is beyond the scope of our study to provide an in-depth critical assessment of this enormous body of literature, we emphasize that our use of the planetary boundaries to define the Doughnut's ecological ceiling is not taken lightly or uncritically—it is an intentional choice based on our understanding of the state of the art in Earth-systems science and, crucially, our aligned commitment to iteratively incorporate new knowledge as that science continually advances. We acknowledge that the limitations of the planetary boundaries framework could affect our results and may require further investigation.

Furthermore, we note that several of the terms used in the planetary boundaries framework have been altered in the Doughnut with the aims to: (1) align terminology across the social and ecological domains (that is, we generally refer to 'dimensions' and 'indicators' throughout, rather than 'Earth-system processes' and 'control variables') and (2) make them more accessible to a non-technical audience.

To establish the ecological ceiling, we began with the list of ecological indicators in the 2023 planetary boundaries update and assessed

their applicability considering our ambition to monitor time series according to two criteria, namely, that ecological indicators should:
- Be part of the 2023 update to the planetary boundaries framework, tracking Earth-system processes in the same units, ideally with the same data sources.
- Be measured with sufficiently recent published data, with global coverage and with time-series monitoring.

We considered dozens of indicators in published data sources from international databases and the scientific literature, reached out to relevant experts and used our own expertise to narrow down a draft set of ecological indicators that most closely matched these criteria for the nine ecological dimensions of the planetary boundaries. We conducted an internal review of this draft set of dimensions and indicators with colleagues holding a range of perspectives: some with insight on planetary boundaries science, others with a focus on specific indicators and data sources (see 'Acknowledgements'). We are grateful for the comments and suggestions shared by colleagues but we do not claim that the internal reviewers cover a representative or exhaustive set of relevant expertise nor do we claim that all reviewers would agree with all of the indicators we have chosen. We take full responsibility for the final indicator selection, as shown in Table 2.

Notably, the chemical pollution indicator is an exception to the criteria above, for which we propose to focus on the production of hazardous chemicals. This metric is distinct from but related to the (unquantified) indicator in the planetary boundaries framework on the 'percentage of synthetic chemicals released to the environment without adequate safety testing'[13]. Our approach combines a focus on the total production of chemicals, which is in line with the landmark study of Persson et al.[56] on novel entities, with calls for nature-based chemicals that are conducive to life, rather than hazardous to health and environment[57,58]. See Supplementary Discussion 2 for further details on each ecological indicator and Supplementary Table 2 for data sources.

## Calculating social shortfall and ecological overshoot

In our Analysis, social indicators are presented relative to their extent below the social foundation and ecological indicators are presented relative to their extent beyond the ecological ceiling. For the social indicators, each one is already expressed in percentage terms as the proportion of the global population falling below its respective minimum standard. These percentages can therefore be interpreted directly as the normalized extent of social shortfall between 0% (no shortfall) and 100% (complete shortfall) each year.

For the ecological indicators, which are expressed in absolute units, the normalization procedure scales each indicator–boundary pair by assigning the pre-industrial Holocene baseline a value of zero, divides each indicator value by its respective planetary boundary and then subtracts one from each normalized ratio (see Supplementary Table 2 for pre-industrial Holocene baselines provided by the planetary boundaries framework[13]). In percentage terms, the normalized extent of ecological overshoot has a lower bound of −100% (the pre-industrial baseline), with 0% indicating no overshoot, and no upper bound for values greater than zero.

In mathematical terms, the general formula for the normalized ecological overshoot in a given year is given by $\text{overshoot}_t = (x_t - x_{\text{base}})/(x^* - x_{\text{base}}) - 1$, in which $x_t$ is the ecological indicator in year $t$, $x^*$ is the planetary boundary and $x_{\text{base}}$ is the pre-industrial baseline. However, three indicator–boundary pairs are exceptions to this general formula (aragonite saturation state, forest area and stratospheric ozone), as they are each framed inversely, for which a decrease in the indicator value implies worsening ecological conditions (rather than an increase). The normalization formula to express overshoot in comparable terms for these inverted indicator–boundary pairs is given by $\text{overshoot}_t = \left(1 - \frac{x_t}{x_{\text{base}}}\right) \Big/ \left(1 - \frac{x^*}{x_{\text{base}}}\right) - 1$.

## Determining historical and scenario trends

We estimated historical trends over time for each indicator of social shortfall and ecological overshoot using ordinary least squares regression and two-sided hypothesis tests with a linear model, or $y = \beta_0 + \beta_1 t$, in which $t$ is a year index with base year 2000, $y$ is the indicator and $\beta_0$ and $\beta_1$ are the regression coefficients (intercept and slope, respectively). See Supplementary Data for the full regression model results for each indicator, including estimated coefficients, robust (heteroskedasticity-consistent and autocorrelation-consistent) standard errors, $P$-values and coefficients of determination (adjusted-$R^2$).

We derived simple indicator-specific scenario pathways to eliminate social shortfall by 2030 and ecological overshoot by 2050 by calculating the annual linear rate of change between 2022 levels and zero for each indicator over its respective period (that is, 8 years for social indicators, 28 years for ecological indicators). We derived these simple scenario pathways to offer a rough sense of the level of ambition required to live within the Doughnut by mid-century, in comparison with the historical trends estimated empirically. Although social shortfall trends and ecological overshoot trends are both expressed in percentage points per year, we note that the values are not directly comparable across the social and ecological domains because they are scaled differently: the former are expressed in a 0–100 scale, whereas the latter are scaled with respect to pre-industrial Holocene baselines that are defined differently for different boundaries.

The illustrative aspiration to eliminate social shortfall in the Doughnut by 2030 is broadly aligned with the social ambition of the SDGs, which all UN member states have committed to achieving by 2030. Unlike the social indicators, we opted against a 2030 aspiration to eliminate ecological overshoot in the Doughnut because there is minimal political support to achieve the specific ecological targets defined by the planetary boundaries framework by this date. Instead, we derived an illustrative 2050 aspiration that is consistent with the political ambition to achieve 'net-zero' greenhouse gas emissions by this date[59] and is broadly aligned with a 50% probability of keeping global heating below 1.5 °C (ref. 60)—a level that reduces the risk of triggering tipping points in the Earth-system, such as major ice-sheet collapse and near-complete coral mortality (although the possibility of crossing such thresholds cannot be ruled out above 1 °C heating, which has already occurred)[61].

## Disaggregating the global Doughnut

The global Doughnut was disaggregated into three country clusters based on average levels of annual gross national income (GNI) per capita over the 2000–2022 period, using data collected from the Human Development Report[62] from the United Nations Development Programme (UNDP), available for 193 countries. GNI per capita is expressed in international dollars (Int-$) at 2017 prices, which means that national values are adjusted for inflation and for differences in living costs between countries. We defined country clusters by income percentile thresholds as follows: poorest 40% of countries (less than Int-$ 8,100 per capita; $N = 78$), middle 40% of countries (between Int-$ 8,100 and 33,200 per capita; $N = 77$) and richest 20% of countries (more than Int-$ 33,200 per capita; $N = 38$). See Extended Data Fig. 7 for country-specific details.

For the social foundation, we calculated the proportion of the population falling below minimum social standards within each country cluster using the same indicators, methods and data sources as the global Doughnut. As such, levels of social shortfall are directly comparable across the global and country-cluster scales. We use a consistent set of 193 countries to define each country cluster but not all countries have data available for all indicators (as noted in more detail in the 'Global time-series data' subsection). We analysed social shortfall by country cluster in the year 2017 to enable comparison with the national environmental footprint data collected from ref. 15, which is only available for this single year (except for the public transport indicator, for which we include 2020 values owing to a lack of earlier data).

For the ecological ceiling, the planetary boundaries are related to critical Earth-system processes, which cannot be disaggregated to smaller scales in a directly comparable manner. However, because the original planetary boundaries framework was proposed in 2009, scholars have been developing methods to translate these Earth-system indicators into finite global resource budgets informed by planetary boundaries science, which can be allocated to individual countries according to a sharing principle. After more than a decade of applied research, a general translation procedure has been established with well-known limitations, acknowledgement of uncertainties and discussion of ethical implications of distinct sharing principles, among other themes[38].

To translate the global Doughnut's ecological ceiling to the country-cluster scale, we collected equality-based per capita boundaries for the year 2017 from a recently published study[15], which are related to four of the Doughnut's ecological dimensions (climate change, nutrient pollution, freshwater disruption and biodiversity breakdown). To compare country-cluster performance with respect to these per capita boundaries, we collected national data for six consumption-based environmental footprint indicators available for 168 countries in 2017 from the same source[15] (carbon dioxide, phosphorus, nitrogen, blue water, species loss and HANPP) and calculated a population-weighted average of each national per capita environmental footprint within each country cluster. The environmental footprints are calculated using input–output analysis, which accounts for spillovers and outsourcing of upstream environmental burdens across countries enabled by international trade by allocating them to final consumers, no matter where in the world such burdens occur. Such spillovers and outsourcing are critical to account for inequalities in consumption across societies but they are not relevant in the global Doughnut, as production and consumption are equal at the global scale.

Ecological overshoot was calculated for each per capita indicator–boundary pair using the same method described above for the global Doughnut (see 'Calculating social shortfall and ecological overshoot' subsection). The planetary boundaries for nutrient pollution and biodiversity breakdown are both represented by two separate indicators (phosphorus and nitrogen, and species loss and HANPP, respectively). We note that this approach to measuring ecological overshoot on the basis of country-cluster per capita consumption with respect to downscaled planetary boundaries should be seen as complementary to assessments of locally relevant ecological pressures and thresholds. Although increasing consumption and affluence, including a growing middle class, are widely held to be the primary drivers of global impacts on Earth-system stability[33], local ecological concerns may be more strongly affected by other factors, such as overexploitation, urban and agricultural encroachment, pollution and population growth[63]. Further details on the individual per capita indicator–boundary pairs are provided in Supplementary Discussion 2 and Supplementary Table 2.

### Calculating proportions of global totals

In our Analysis, we present the share of global population in social shortfall held by each country cluster in 2017 alongside the contribution of each country cluster to global ecological overshoot in the same year (see Extended Data Fig. 10). In both cases, the results are expressed as proportions of total shortfall and total overshoot in 2017 in percentage terms.

For each social indicator, we calculated the number of people in deprivation in each country cluster by multiplying its proportion of population in deprivation by its total population. These country-cluster populations in deprivation were summed to give total global population in shortfall and the share of global social shortfall held by each country cluster was defined as its respective proportion of this total population, or in mathematical terms for each country cluster $n$: share of global shortfall$_n$ = (share deprived$_n$ × population$_n$)/population deprived$_{total}$.

For each ecological indicator, we calculated the absolute level of environmental footprint beyond equality-based population shares in each country cluster by subtracting the global per capita boundary from its per capita environmental footprint and then multiplying this excess environmental footprint per capita by the country cluster's total population. These excess environmental footprints per country cluster were summed to give total environmental footprint overshoot and the contribution of each country cluster $n$ was defined as its respective proportion of this total overshoot, or in mathematical terms: share of global excess EF$_n$ = ((EF per cap$_n$ − boundary per cap$_n$) × population$_n$)/excess EF$_{total}$, in which EF is the environmental footprint indicator.

We note that this single-year approach does not account for the historical contributions of wealthy countries to global ecological overshoot, which is not ideal but compiling comparable national time-series data on environmental footprints and per capita boundaries from disparate sources is a non-trivial exercise that was beyond the scope of the present study. Building on previous research showing that wealthy countries contribute approximately 90% of excess cumulative $CO_2$ emissions beyond population-based fair shares of safe global carbon budgets (compared with 50% using our single-year approach)[64,65], a cumulative assessment of trends in ecological overshoot over time for several footprint indicators across country clusters is an important avenue for future research, in our view.

### Limitations

Our Analysis is necessarily limited by the quality and availability of global time-series data (detailed descriptions of each social and ecological indicator are provided in Supplementary Information). Furthermore, although the Doughnut monitoring framework tracks each social and ecological indicator separately, we acknowledge the interdependence of many indicators. Although we do not analyse such complex interdependencies formally here, the representation of indicators in the Doughnut diagram conveys a visual sense of holistic interconnection that could frame and support future research in this area, which is beginning to emerge[26,48,49]. That being said, we note that the radial representation of indicators, such as the Doughnut plots in Figs. 1 and 3 (and the planetary boundaries framework), has been criticized, notably because quantitative values scaled in terms of wedge radius leads to a quadratic increase in wedge area[66]. To mitigate the risk that some readers may perceive small changes as more significant than they are, we also represent comparable results using bar charts (Figs. 2 and 4).

We acknowledge that our focus on the global and country-cluster scales masks wide inequalities in levels of social shortfall and ecological overshoot between countries and within them. We recognize the importance of accounting for such inequalities, particularly to envision equitable trajectories towards the Doughnut, that we see as complementary to our approach monitoring high-level social and ecological trends on a changing planet, which also need to be taken into account for considering Earth-system trajectories. We note that the environmental footprint data from ref. 15 is also available by expenditure decile within each country, but we analysed national averages to enable comparison with the social indicators, which are not generally available with such within-country disaggregation. A useful step for future research could therefore be to account for within-country social deprivations by income groups and/or other characteristics in a cross-country comparable framework (see ref. 67 for a single-country application in Norway).

Finally, the Doughnut of social and planetary boundaries will continue to evolve. Its social foundation—including the dimensions, indicators, boundaries and data—will continue to be revised as internationally agreed social norms and standards continue to evolve and as improved international data become available. We acknowledge that some of the indicator thresholds that we have selected in this iteration lack officially recognized thresholds of minimum acceptable standards at present, such as a homicide rate of 5 or more per 100,000, whereas other

indicators lack data altogether, such as racial equality and several disag-gregated ecological boundaries. However, in these relatively early days of devising metrics fit for monitoring progress with respect to social and ecological goals, our view is that such data gaps are to be expected and one of the best ways to improve them is by making them visible. Future iterations of the social foundation could, for example, include dimensions concerning cultural rights and community resilience. Like-wise, the ecological ceiling's dimensions, indicators, boundaries and data will continue to be revised and refined as scientific research and understanding of Earth-system processes proceeds and gets translated for application at smaller scales. Future iterations could, for example, include more specific forms of chemical pollution, such as plastics, and improved metrics for air pollution and biosphere breakdown.

## Data availability

Data sources for each social indicator are described in 'Supplementary Discussion 1 – The social foundation: dimensions and indicators' and summarized in Supplementary Table 1. Data sources for each ecological indicator are described in 'Supplementary Discussion 2 – The ecological ceiling: dimensions and indicators' and summarized in Supplementary Table 2. The data produced in this study are included in the Supplementary Data spreadsheet accompanying this article and are also archived on Zenodo (v1.0.1) at https://doi.org/10.5281/zenodo.15688961 (ref. 68). The data are also available to explore through an interactive webpage (https://doughnuteconomics.org/doughnut/) that allows users to query the dataset and visualize doughnut plots similar to Figs. 1 and 3 showing social shortfall and ecological overshoot over time.

## Code availability

The data analysis was conducted using R (v4.4.1). Beyond this base R version, our analysis is dependent on several R packages. We used the tidyverse suite of packages (v2.0.0) for organizing, manipulat-ing and visualizing the data. We also used the zoo package (v1.8-12) for time-series analysis functionality, the lmtest package (v0.9.40) and the sandwich package (v3.1.1) for statistical analysis, the jsonlite package (v1.8.9) to convert vector data to nested json format and the ggpubr package (v0.6.0) for further data visualization functionality. The doughnut plots were rendered on the Observable platform using the D3 (v7.0.0) javascript library. The source data and custom R code used to generate the analysis are archived on Zenodo (v1.0.1) at https://doi.org/10.5281/zenodo.15688961 (ref. 68).

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

**Acknowledgements** We thank R. Priester for graphic design support and we are grateful to the many colleagues who shared time and expertise with us during the data collection and internal review processes, including: A. Beusen, A. Bjørn, A. Lavaquial, A. Whitten, C. A. Fergus, D. Joliffe, D. O'Neill, D. Sullivan, H. Haberl, J. Hickel, J. Mahmood, K. K. Pal, K. Richardson, M. Moatsos, M. Porkka, M. Roser, M. Rounsevelle, M. Weber, P. Antrobus, P. Tian, R. Geyer, R. Salawitch, S. Cornell, V. Songwe and Y. Iida.

**Author contributions** A.L.F. and K.R. conceptualized and designed the study. A.L.F. contributed data management and visualization. A.L.F. and K.R. contributed analysis, writing and editing.

**Competing interests** The authors declare no competing interests.

**Additional information**
**Correspondence and requests for materials** should be addressed to Andrew L. Fanning.

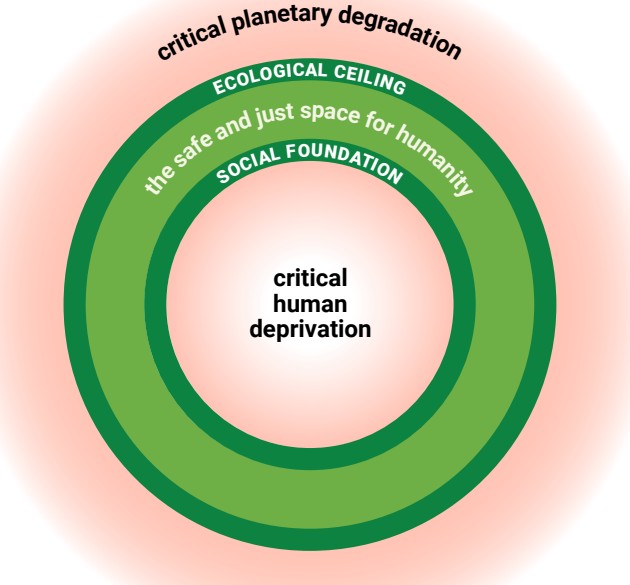

**Extended Data Fig. 1 | The core doughnut-shaped conceptual framework.** Adapted from ref. 2.

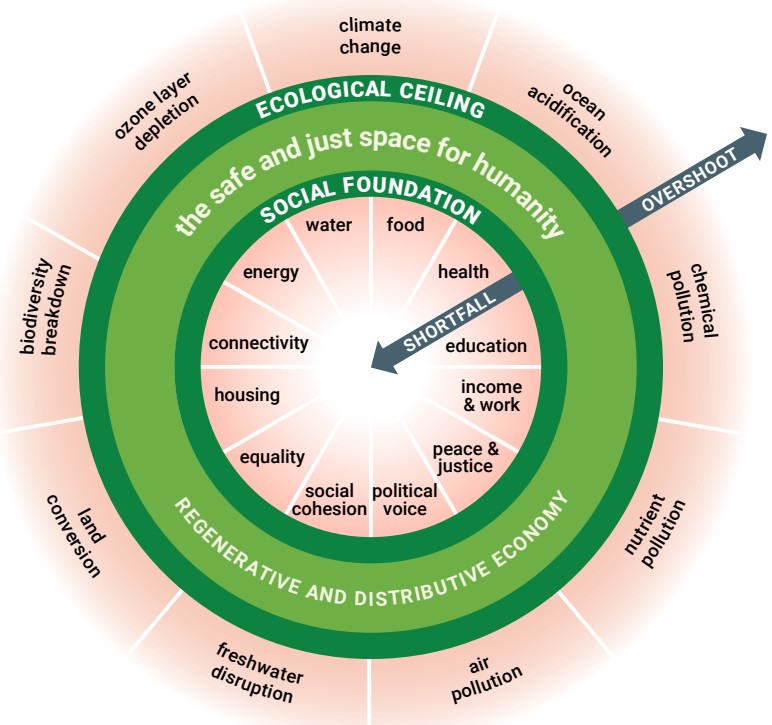

**Extended Data Fig. 2 | The third iteration of the Doughnut of social and planetary boundaries.** See ref. 52 for a comparison with previous iterations of the Doughnut. Adapted from ref. 2.

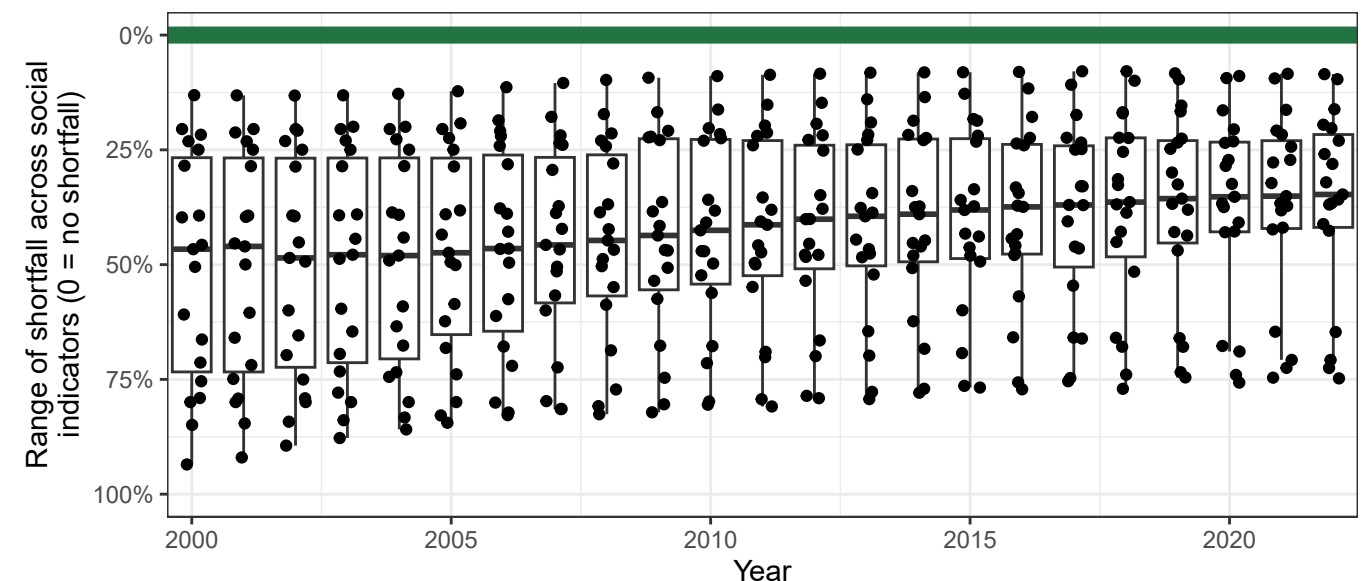

**Extended Data Fig. 3 | Range of social shortfall across global indicators by year (*N* = 19).** Data points are individual observations for 19 social indicators with available time-series data in each year (with jitter to avoid overlaps). Each social indicator tracks the share of the world's population falling below its respective minimum social standard (green line) in percentage terms, ranging from zero (nobody in shortfall) to 100 (entire population in shortfall). For each box plot, the horizontal line and the box represent the median and the interquartile range (IQR), respectively, and the 'whiskers' extend to the closest observations greater than/less than 1.5 × IQR. To ensure a consistent set of indicators over time, we exclude two indicators with insufficient time-series data (food insecurity, which starts in 2015, and lack of public transport, which starts in 2020) and we fill observations for three indicators missing values between 2000 and 2004 by carrying their respective 2005 observations backward (youth NEET, lack of social support and perceptions of corruption).

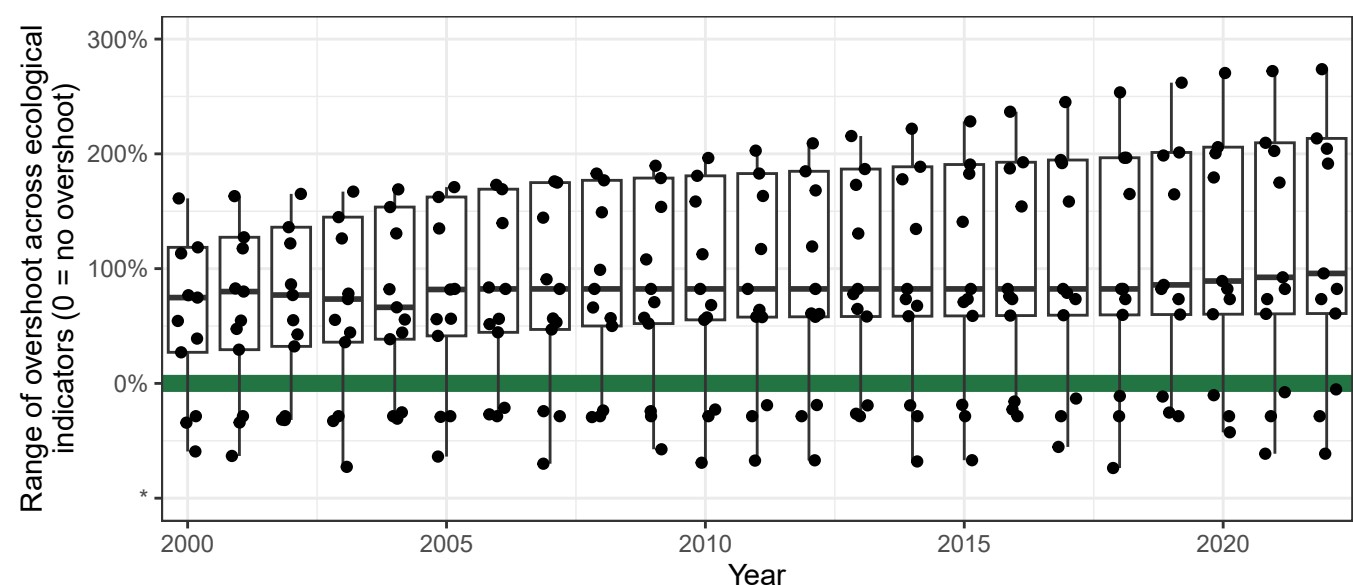

**Extended Data Fig. 4 | Range of ecological overshoot across global indicators by year (N = 13).** Data points are individual observations for the ecological indicators in each year (with jitter to avoid overlaps). Each ecological indicator tracks the Earth-system status of its respective planetary boundary in percentage terms, for which zero indicates no overshoot and ranging from the pre-industrial Holocene baseline (*) to an unbounded upper extent of overshoot). For each box plot, the horizontal line and the box represent the median and the interquartile range (IQR) respectively, and the 'whiskers' extend to the closest observations greater than/less than 1.5 × IQR. Outliers for extinction rate and hazardous chemicals extend beyond the chart area (900% and 1,410–3,210% depending on the year, respectively).

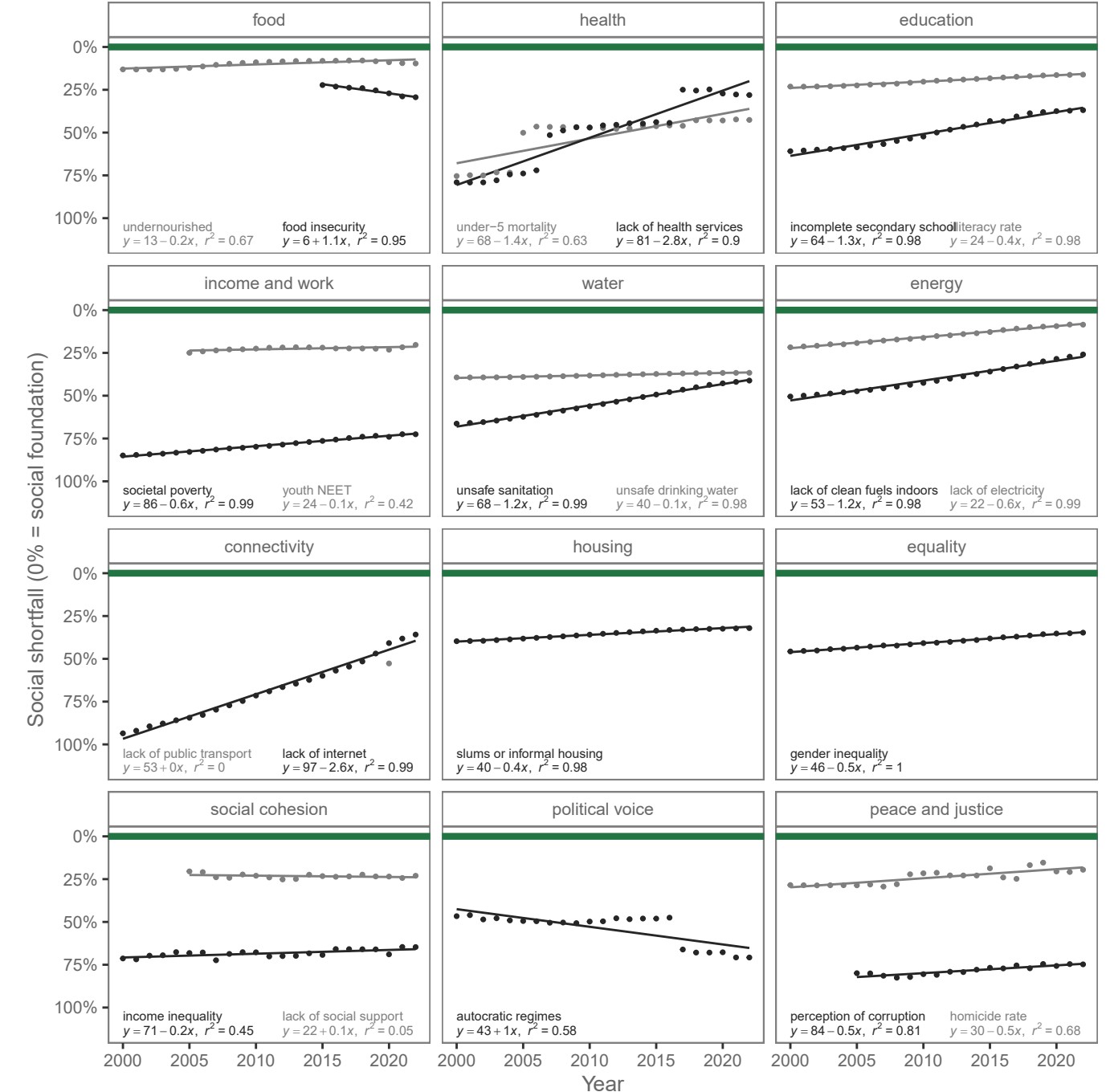

**Extended Data Fig. 5 | Historical observations and trends across the global Doughnut's social dimensions and indicators of shortfall.** Each social indicator tracks the share of the world's population falling below its respective minimum social standard (green line) in percentage terms, ranging from zero (nobody in shortfall) to 100 (entire population in shortfall). See Table 1 for further details on the indicators and Supplementary Data for numerical results, including full statistical details estimated using ordinary least squares regression with two-sided hypothesis tests (that is, estimated coefficients, robust standard errors, adjusted-$R^2$ and $P$-values).

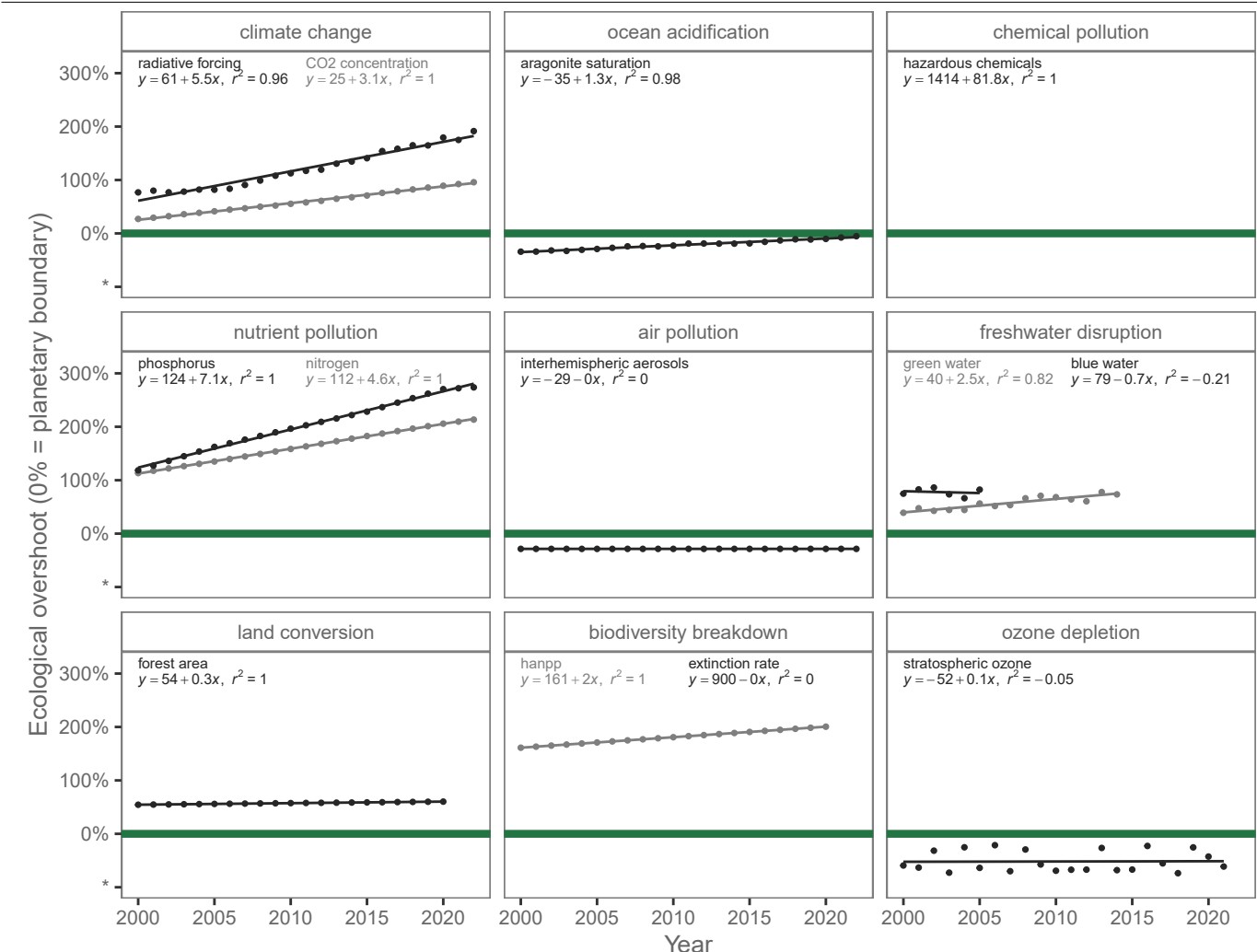

**Extended Data Fig. 6 | Historical observations and trends across the global Doughnut's ecological dimensions and indicators of overshoot.** Each ecological indicator tracks the Earth-system status of its respective planetary boundary (green line) in percentage terms, for which zero indicates no overshoot, and ranging from its pre-industrial Holocene baseline (*) to an unbounded upper extent of overshoot. See Table 2 for further details on the indicators and Supplementary Data for numerical results, including full statistical details estimated using ordinary least squares regression with two-sided hypothesis tests (that is, estimates, robust standard errors, adjusted-$R^2$ and $P$-values).

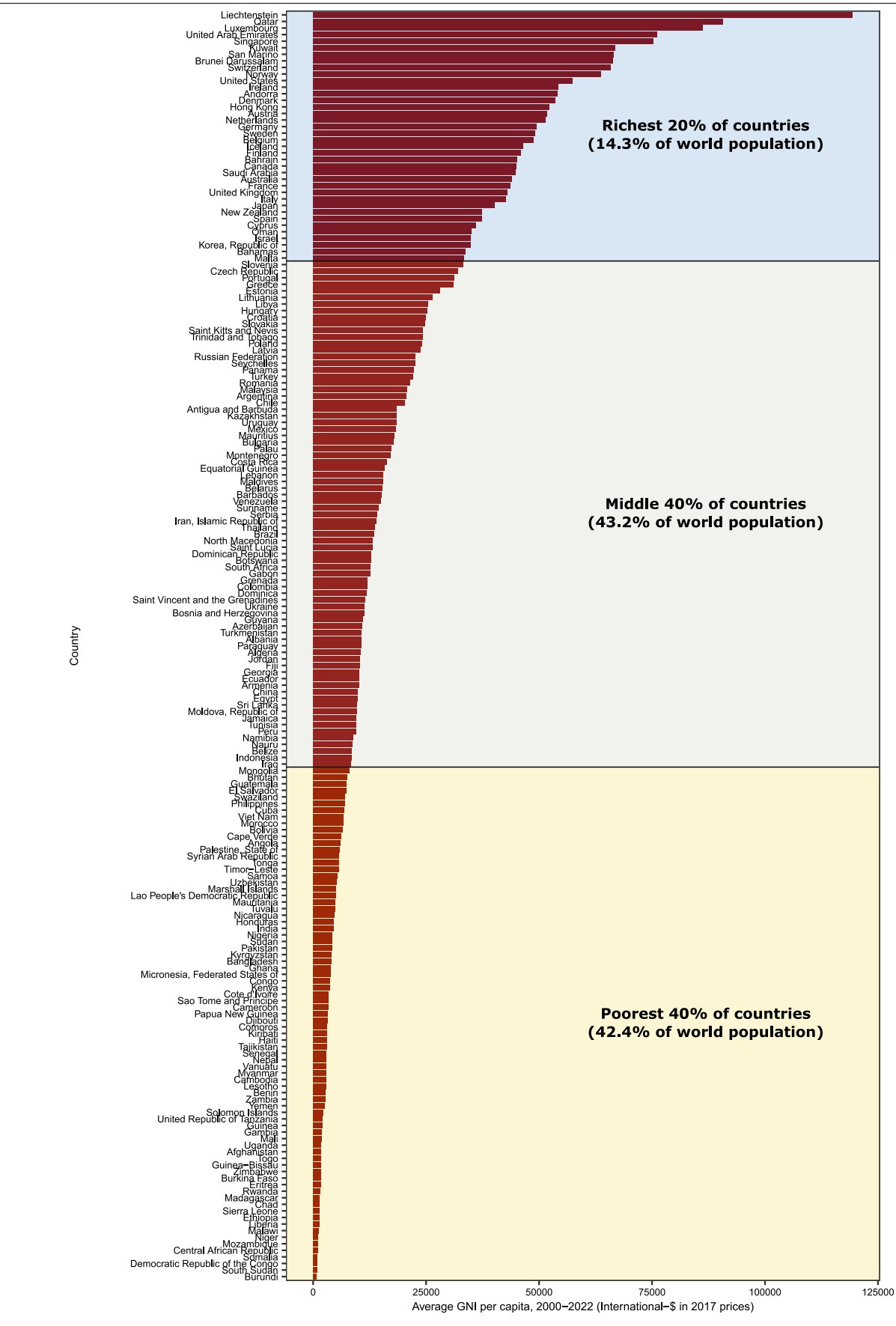

**Extended Data Fig. 7 | Country-specific income per capita in the three country clusters analysed in this study.** Countries are ranked by GNI per capita, adjusted for inflation and for differences in living costs (Int-$ at 2017 prices), averaged over the 2000–2022 period (*N* = 193).

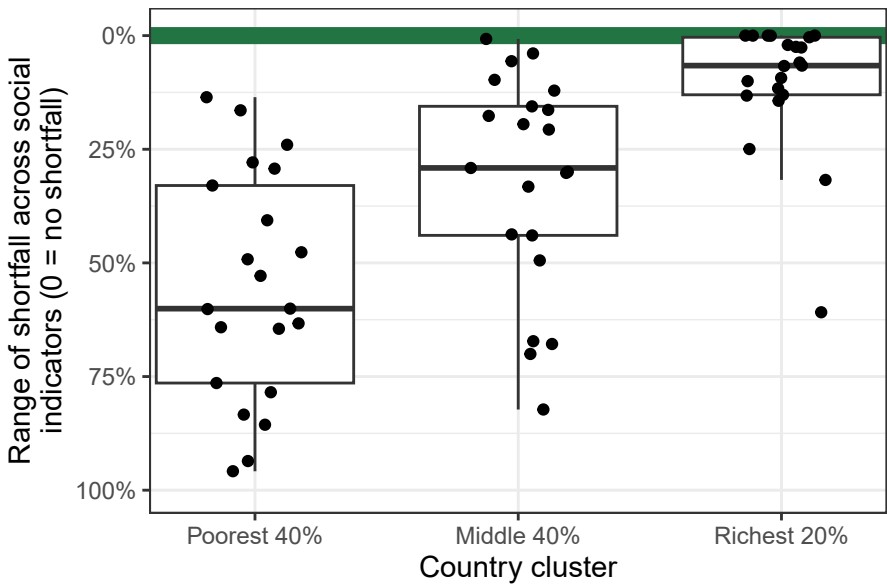

**Extended Data Fig. 8 | Range of social shortfall across disaggregated indicators by country cluster (*N* = 21).** Data points are individual observations for the 20 social indicators with available data in 2017 (plus the public transport indicator, which shows 2020 values owing to a lack of earlier data), with jitter to avoid overlaps. Each social indicator tracks the share of the population falling below its respective minimum social standard (green line) in percentage terms, ranging from zero (no shortfall) to 100 (entire population in shortfall). For each country-cluster box plot, the horizontal line and the box represent the median and the interquartile range (IQR) respectively, and the 'whiskers' extend to the closest observations greater than/less than 1.5 × IQR.

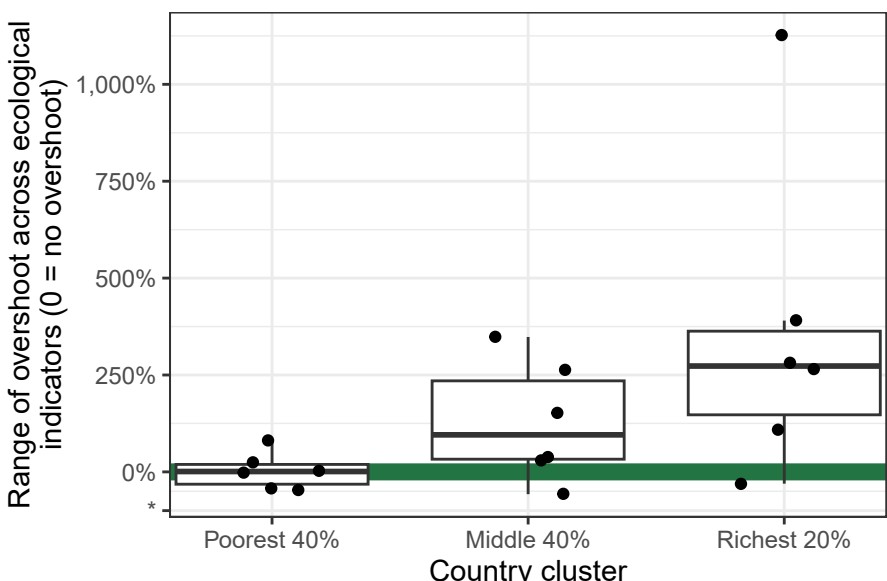

**Extended Data Fig. 9 | Range of ecological overshoot across disaggregated indicators by country cluster (*N* = 6).** Data points are individual observations for the ecological indicators with available data in 2017 (with jitter to avoid overlaps). Each ecological indicator tracks a consumption-based environmental 'footprint' relative to its respective downscaled per capita boundary (green line) in percentage terms, for which zero indicates no overshoot, and ranging from no environmental footprint (*) to an unbounded upper extent of overshoot. For each country-cluster box plot, the horizontal line and the box represent the median and the interquartile range (IQR) respectively, and the 'whiskers' extend to the closest observations greater than/less than 1.5 × IQR.

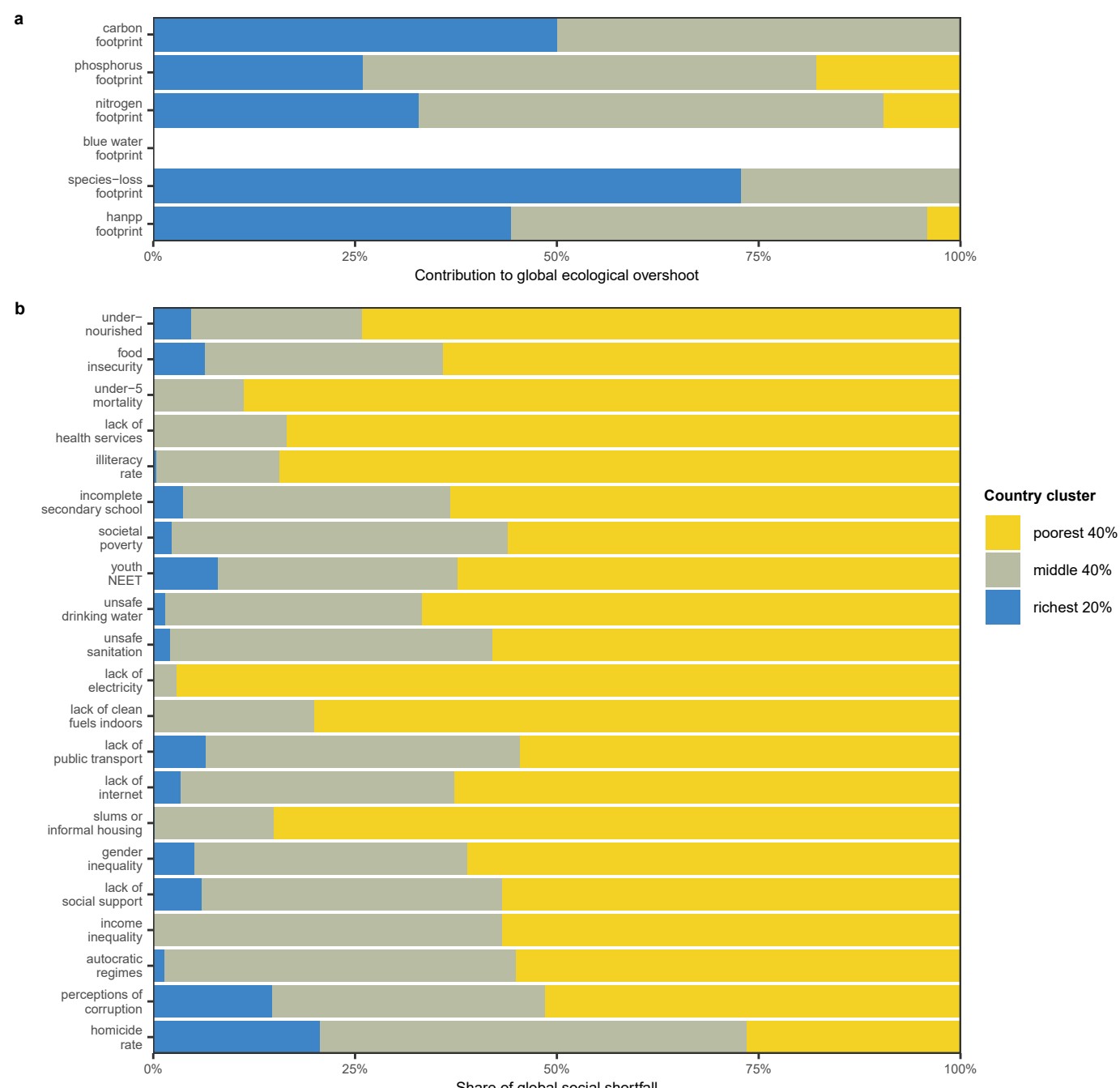

**Extended Data Fig. 10 | Shares of global social shortfall and global ecological overshoot by country cluster. a**, Relative contributions of country clusters to global ecological overshoot, measured as the proportion of total excess environmental footprint held by each country cluster beyond its population share of per capita boundaries. **b**, Relative shares of global social shortfall across country clusters, measured as the proportion of total population falling short of minimum social standards held by each country cluster.