## [Peer Review file · Nature]

Doughnut of social and planetary boundaries monitors a world out of balance

Corresponding Author: Dr Andrew Fanning

Version 1:

Reviewer comments:

Referee #1

(Remarks to the Author)

This paper provides a first temporal assessment of whether the world is overshooting ecological boundaries and falling short of a social foundation. The paper does so with a carefully considered set of ecological and social dimensions and indicators from various data sources. The paper gives a clear picture of the world being on a dangerous path of increasingly overshooting ecological boundaries with comparatively small gains towards the social foundation. The paper is well written, the method is well explained, and the topic is certainly of relevance and interest to readers of Nature. I have three main comments and some minor points.

Main comments

1. Contribution: I think the contribution falls short of the bar of Nature, and recommend the authors add more value by breaking down the results at the country level.

I understand the contribution of the paper to be a revision of the doughnut framework and an application of the framework to temporal analysis. I find both impressive and relevant, yet in my understanding, this contribution falls short of the more groundbreaking contributions appearing in Nature (a call for the editor to make, to be sure). I certainly understand that revising the indicators is not as simple as it may sound, and one of the impressive aspects of this paper is to take something that is not simple at all and make it digestible for a general audience. Yet, I would like to see a greater contribution, possibly by extracting more relevant conclusions from the data. Most notably, it should be possible to say something at the country level about the social shortfall and ecological overshoot. Most of the indicators in the social foundation are already available at the country level, and even if some of these indicators have missing entries, those could be imputed. Less of the ecological indicators are available at the country level, yet the ones that are not could partially be computed at the country level using studies such as this one: <https://doi.org/10.1038/s41597-023-02041-1>. Even if it is ultimately not possible to generate country level estimates for all indicators, it is possible for sufficiently many to say something about which countries are responsible for overshooting the ecological boundary, which countries have been able to make progress on the social foundation at the smallest ecological cost etc.

2. Use of relative and subjective indicators: I recommend that the authors remove relative indicators and strongly caveat subjective indicators.

The indicators under the social foundation are partly subjective and relative, both of which I find inappropriate for this exercise, and both of which are partially driving the results. Most of the indicators with small or reverse progress in the social foundation fall under this category. I think the relative indicators must be replaced, revised, or removed and the subjective indicators either the same or heavily caveated. My understanding of the Voice and Accountability Index and the Control of Corruption Index is that they rely on a relative scale. This means that even if all countries make substantive progress, the global shortfall in these indicators need not fall. The authors should either reconstruct these indicators such that they

measure absolute progress or replace them altogether. The subjective indicators (life satisfaction and food insecurity) can also be problematic. For example, the Food Insecurity Experience Scale has been found to very weakly relate to objective measures of food insecurity (<https://doi.org/10.1093/wber/lhad031>), casting doubt on what exactly it captures. Likewise, it is well known that individuals (and cultures) interpret the life satisfaction scales differently and that any conclusions from such measures may not be robust to alternative understanding of the scales (see here an application to happiness, not life satisfaction: <https://www.journals.uchicago.edu/doi/full/10.1086/701679?mobileUi=0>). To be sure, FIES and life satisfaction also pick up relevant and important parts of a social foundation, but the measures could certainly be caveated more by the authors.

3. Distinction between inequality and social cohesion: I recommend that the authors more clearly delineate these two dimensions and rethink the indicators underpinning them.

Both the inequality dimension and social cohesion dimension talk about inequality of opportunity. Somewhat strangely, the only measure of inequality of outcomes (the Palma ratio) is not in the inequality dimension but in the social cohesion dimension. The other indicator under social cohesion, life satisfaction, surely is related to social cohesion, but life satisfaction is also related to all other dimensions tracked. I don't think the authors sufficiently argue for its inclusion and I think the prior indicator used (self-reported social support) seems to be closer to the concept the authors are after. Sure, it reduces the complexity of a respondent's close relationship into a binary yes/no, yet the entire exercise in the social foundation is about reducing complex relationships to binary states. My recommendation would be to move the Palma Ratio to the equality dimension and replace life satisfaction with something more closely tied to social cohesion.

Smaller comments

The doughnut framework is a powerful concept for thinking about a social foundation and ecological ceiling, yet as a visualization tool for presenting data, it has an important flaw if drawn the way in Figure 1. The first sentence of the Wikipedia entry on Pie charts makes the point better than I can: "A pie chart (or a circle chart) is a circular statistical graph which is divided into slices to illustrate *numerical proportion*" [I added the asterisks]. As far as I understand, the areas in Figure 1 do not match numerical proportions, rather, one gets the visual impression that the overshoots are disproportionately larger than the undershoots. To be sure, this is not a pie chart, but I do think people seeing this chart intuit that it illustrates numerical proportions. This issue arises, I think, because the authors match the *height* of the areas in Figure 1 with the shortfall/overshoot, and not the *areas* of Figure 1 with the shortfall/overshoot. Solutions to this issue could be to present Figure 1 as Figure 2 or to mention in the figure note that the areas are not proportional to actual shortfall/overshoot.

In Table 1 and 2 where it says "% per year", should it be percentage points? Similarly, in discussions of those two tables, I think "%" is often used instead of percentage points.

What explains the large jump in under-5 mortality around 2005 in Figure 3? It seems like something is off with the data. I understand the authors want to be data takers, yet in this case, if some correction can be made, this would increase the credibility of the findings. This large jump does not appear in the WDI: <https://data.worldbank.org/indicator/SH.DYN.MORT?end=2022&start=1990&view=chart>

Under Acknowledgments, I think it would be appropriate to name the experts who provided feedback to the draft set of dimensions and indicators.

The authors mention that a structural weakness of using broad-gauge purchasing power to monitor poverty is that countries may have different prices for essential goods. Though certainly true in theory, in practice, this concern does not appear to have much merit: <https://www.aeaweb.org/articles?id=10.1257/app.3.2.137>.

I appreciate the desire to track racial equality. My understanding is that in some countries, the type of inequality the authors seek to measure is better described as ethnic inequality. Perhaps the indicator name could be changed to reflect both types of inequalities (as indeed does the discussion in the supplementary information).

The authors write that they "collected data on self-reported life satisfaction." I doubt they actually collected the data themselves, so perhaps the sentence could be rephrased to avoid confusion.

Perhaps one sentence could be added explaining how human-induced radiative forcing complements the CO2 concentration. What does the former capture that the latter does not? I am not doubting the choice, only suggesting that one more sentence could be helpful for some reasons. Likewise, I recommend adding a sentence on how 350 ppm CO2 relates to the Paris agreement and GMST increases.

The authors mention that the indicators and dimensions used are related, and though this definitely will be impossible (and probably also undesirable) to avoid, in at least one case, the indicators are not just related but overlapping: Access to improved water and sanitation is part of the housing indicator and its own indicator. What about removing this part from the housing indicator?

(Remarks on code availability)

Referee #2

(Remarks to the Author)

Review of “Doughnut of social and planetary boundaries monitors a world deeply out of balance”

This paper essentially reports on a data update of the Doughnut framework. It reports on an extension of indicators, and expansion of the data foundation to a time series. The Doughnut framework has attracted significant international attention, so its update is significant in itself.

General comments:

- I enjoyed reading the manuscript; my main comment is around the going forward section. Here, the first paragraph is a summary of the preceding main text. The second paragraph talks about varying capacities around the world. The third mentions “tools of doughnut economics”, but then settles on annual updates to be provided. I would have hoped to read a bit more about the way forward. Now that we have understood the dire situation, what do we do with your updated data? How – in concrete terms – can it help us better in changing our current predicament? What tools exactly would you suggest that researchers use in combination with your database, and how would they use them? What new and updated research directions emerge from your new data? Some insights would be great.

- A comparison with similar time series data sets would be great. I’m sure there are a few, but I’d mention one here, UNEP’s Sustainable Production and Consumption Hotspots Analysis Tool (SCP-HAT; <https://scp-hat.org>) is one of them. Also here, you find time series, and many environmental, resource and social indicators. Also this database is being regularly updated, under UNEP’s auspices. What can we learn from the different frameworks? How does the Doughnut stand out?

- I mention SCP-HAT, because it includes spill-overs and outsourcing, which I believe the Doughnut does not. This is a major factor in monitoring SDGs. Affluent countries outsource dirty and unfair production to regions with lax regulations, mostly in the global south. This outsourcing contributes majorly to countries’ – and the world’s – performance on many indicators. See for example a recent publication on polarising trends on multiple indicators¹, which I believe speaks very well to your observations, but adds some analytical rigour. Outsourcing effects are also explicitly reflected for example in SDGs 8 and 12, through the material footprint indicator², which is monitored by the SCP-HAT website.

Specific comments:

- This is a minor comment. You write: “static snapshot” and “dynamic monitor” when you refer to your time series. In my understanding, the term “dynamic” implies some sort of intertemporal interaction. What you have is a series of static snapshots, but that does not make the time series “dynamic”, as in the sense that a previous year acts on the following year by some sort of acceleration or retarding. I realise this might be word splitting, but I thought I mention it nevertheless. A truly multi-indicator dynamic analysis on the SDGs was recently published, using CGE analysis.³

- You state that “we acknowledge the interdependency of many indicators. While we do not analyse such complex interdependencies formally here...” because the radar diagram “conveys a visual sense of holistic interconnection”. However, one could argue that in portraying (visually) a suite of indicators without consideration of interconnection might provide a distorted message. Imagine I chose to portray environmental and resource issues choosing i) GHG emissions, ii) energy use, iii) deforestation, iv) biodiversity loss and v) ozone layer depletion. Indicators i) and ii), as well as i) and iii), and iii) and iv) are highly correlated, and would yield the same trends, whereas v) is much less correlated to i)-iv). Portraying those would give the sub-suite i)-iv) undue weight. There are methods that deal with criteria overlap, such as the Choquet integral⁴⁻⁶. You might want to consider this, eg for the radar diagram, or in case you ever want to establish country rankings.

Having said this, overall this is a great effort, and will be used and cited a lot.

References

- Malik, A. et al. Polarizing and equalizing trends in international trade and Sustainable Development Goals. *Nature Sustainability*, doi:10.1038/s41893-024-01397-5 (2024).
- Lenzen, M. et al. Implementing the material footprint to measure progress towards Sustainable Development Goals 8 and 12. *Nature Sustainability* 5, 157-166, doi:10.1038/s41893-021-00811-6 (2022).
- Han, S. et al. Prospects for global sustainable development through integrating the environmental impacts of economic activities. *Nature Communications* 15, 8424, doi:10.1038/s41467-024-52854-w (2024).
- Marichal, J.-L. An axiomatic approach of the discrete Choquet integral as a tool to aggregate interacting criteria. *IEEE Transactions on Fuzzy Systems* 8, 800-807 (2000).
- Rowley, H. V., Geschke, A. & Lenzen, M. A practical approach for estimating weights of interacting criteria from profile sets. *Fuzzy Sets and Systems* 272, 70-88, doi:http://dx.doi.org/10.1016/j.fss.2015.01.011 (2015).
- Marichal, J.-L. & Roubens, M. Determination of weights of interacting criteria from a reference set. *European Journal of Operational Research* 124, 641-650 (2000).

(Remarks on code availability)

Referee #3

(Remarks to the Author)

Doughnut economics is a wonderful communication tool to show progress towards or away from sustainable development goals. It frames meeting minimum socio-economic goals while not exceeding environmental limits in an intuitive and compelling manner. This paper updates socio-economic and environmental indicators and offers a few modest extensions of the prior published work on doughnut economics.

The main message of the paper is that global society is seriously off track (“deeply out of balance”) in meeting sustainable development objectives. The paper finds that there have been minor gains in many socio-economic goals but a rapid deterioration in Earth’s life-support systems over the past two decades. These results are consistent with the findings of other recent publications, including the UN Sustainable Development Report, Global Environment Outlook, World Development Report, IPCC AR6, the IPBES Global Assessment, Richardson et al. (2023), Rockström et al. (2009), Steffen et al. (2015), and others. Much of the paper is indeed built from data sources developed by these other assessments (e.g., sustainable development indicators contained in the Sustainable Development Report, and planetary boundary indicators contained in Richardson et al. 2023 and earlier planetary boundary papers). I don’t think the paper offers much that is of interest in addition to what is in these assessments, and I did not learn much that is new from reading the paper. Given this, I don’t think the paper should be published in Nature. The only argument at present I can see for publishing the paper in Nature is that it would provide more of a spotlight on what is a very useful communication tool, one that hasn’t previously been published in a high-profile scientific journal like Nature. Other than this argument, I think the authors would need to revamp the paper in a significant way to provide notable contributions to what already exists in the literature to make the case for publication.

I have a number of relatively specific and mostly minor comments that could be addressed in a revision should the editorial decision be to proceed (or to help the authors in submitting to another journal).

1. The opening sentence could be rewritten to be clearer. The way it is written it seems like you are claiming that there was an assumption of endless growth in GDP, which would be hard to demonstrate, rather than simply stating that growth in GDP was taken to be the predominant measure of success, which could be more easily substantiated.

2. While I think it is accurate to say that growth in GDP was a dominant measure of success since at least the 1950s when measurement of GDP became widespread, there have been counterarguments present from the beginning. Even the economists who pioneered the measurement of GDP knew that GDP was NOT a measure of welfare but simply a measure of income as measured by flows of values in market economies. I also think it is too optimistic to say that society as a whole has outgrown its infatuation with growth in GDP as a measure of success (just look at the rhetoric in almost any national election).

3. Lines 55-62: this paragraph, which is mostly self-promotional, could be deleted.

4. The paper relies on the “planetary boundaries” framework for characterizing ecological overshoot. Like doughnut economics, the planetary boundaries framework is a wonderful for simple communication and is widely referenced. However, the planetary boundaries framework has also been criticized as lacking sound scientific foundations. For example, unlike greenhouse gases that mix in the global atmosphere, excess nitrogen and phosphorus cause problems in specific regions. It doesn’t make logical sense to talk about nitrogen and phosphorus as having a global boundary, even if one derives it by taking averages of regional levels. I think it would be much more informative to talk about the share of regions that exceed critical levels, similar to what is done on the socio-economic side where it is the share of people who live below certain standards. I am also not convinced that HANPP is a meaningful measure of a planetary boundary. It is a measure of the scale of human activity, but does it really indicate an ecological overshoot in the way that loss of biodiversity or increases in greenhouse gas emissions, N and P overloads, or other metrics do? Having some critical assessment of planetary boundaries would add value to the work. Simply presenting the work of Richardson et al. (2023) again does not add much value.

5. On the socio-economic side, the paper uses many of the sustainable development indicators. There are many more sustainable development indicators that are not used. A stronger justification for why you use this set of indicators and not others would strengthen the paper. Some thresholds appear to be fairly arbitrary (e.g., 5 homicides per 100,000 per year). When looking at trends, arbitrary cutoff values are less important, and one could do further analysis to show that the trends are robust to other reasonable (arbitrary) choices of the cutoff value.

6. It is puzzling to me that measures of malnutrition show declining malnutrition while measures of food insecurity are rising. Is there an explanation?

7. In lines 230-231, you state that ecological overshoot “is overwhelmingly being driven by excessive consumption and control over production processes by the affluent.” I think this statement is factually correct for greenhouse gas emissions and several other metrics, but is not clearly true for some others, such as loss of habitat. In recent decades agricultural expansion is happening in low-income countries, driven by rising population and stagnant yields, but not in high-income countries. A close look at the data also shows that rising land use in low-income countries is primarily driven by rising domestic consumption rather than rising net exports of crops to high-income countries.

8. I did not find the exercise of simply projecting past trends into the future to be very informative. However, if you are going to do this, then I would also indicate that the ocean acidification indicator, though currently within its boundary, will cross its boundary by 2050 unless the trend is changed.

9. The final paragraph (lines 237-244) seems more like a sales pitch for the lab than a concluding paragraph. I would largely delete the current paragraph, keeping only the parts about the importance of data for bending the curves on trendlines. It would also be helpful to indicate awareness of the difficulties of overcoming barriers to change needed for successfully bending the curves.

(Remarks on code availability)

Referee #4

(Remarks to the Author)

"The Doughnut" has made a useful contribution to the public debates on social and ecological planetary boundaries. Much of the past scholarship on the Doughnut has eschewed critical reflections about the limitations and ways to address them. This manuscript seems to acknowledge these limitations by hand-waiving at questions of colonialism and inequalities. Yet, the manuscript makes no advances in offering an analytical solution to these gaps and critiques. This would be a missed opportunity.

1) In the spirit of doing good science, I encourage the authors to look for and engage with some critical arguments. More importantly, tackling some of these critiques also offer an opportunity to advance debates on transformational change in a world-ridden by multiple and interconnected crises of wealth inequalities, environmental degradation, and the climate crisis. The following may be a good starting point but authors should undertake a more systematic review of critical takes on the Doughnut: Drees, L., Luetkemeier, R. and Kerber, H., 2021. Necessary or oversimplification? On the strengths and limitations of current assessments to integrate social dimensions in planetary boundaries. *Ecological indicators*, 129, p.108009.

2) By now there is abundant evidence to suggest that global inequalities and ecological degradation are deeply interconnected. Yet, the Doughnut continues to perpetuate this binary between the social foundations and "the humanity's" footprint. While the authors recognize this, they should also draw out implications, and preferably, offer a preliminary empirical analysis of the extent to which global inequalities are responsible for the massive overshoot on ecological indicators. This is especially relevant given the evidence that the burden of addressing the environmental crisis has been imposed on countries with high levels of economic inequalities and missing or dysfunctional democracies. Kashwan, P., 2017. Inequality, democracy, and the environment: A cross-national analysis. *Ecological Economics*, 131, pp.139-151.

3) The IEA data suggesting that 9% of global population lacks access to electricity has proven to be unreliable. See, Min, B., O'Keeffe, Z.P., Abidoye, B., Gaba, K.M., Monroe, T., Stewart, B.P., Baugh, K. and Nuño, B.S.A., 2024. Lost in the dark: A survey of energy poverty from space. *Joule*.

4) The authors might do well to reflect deeply on the extent to which global modeling of these scenarios obscures multi-level complexities and promotes copy-cat aggregate analyses in contexts that differ wildly. Schultz, B., Brockington, D., Coleman, E.A., Djenontin, I., Fischer, H.W., Fleischman, F., Kashwan, P., Marquardt, K., Pfeifer, M., Pritchard, R. and Ramprasad, V., 2022. Recognizing the equity implications of restoration priority maps. *Environmental research letters*, 17(11), p.114019.

5) This is relevant for Global South countries that lack a rule of law framework and quality datasets but also in the Global North countries with extreme inequality: Tønnessen, M. Wasted GDP in the USA. *Humanit Soc Sci Commun* 10, 681 (2023). <https://doi.org/10.1057/s41599-023-02210-y>

(Remarks on code availability)

Referee #5

(Remarks to the Author)

A updated doughnut graph of ecological and social indices

B these indices are not positioned in the large space of climate and social indices. Originality not clear. Only the doughnut presentation seems original, which is science communication rather than original science.

C expert judgment method is under par, experts names and qualifications, method of recruitment, rationales etc not given

D no treatment of uncertainties

F retrievable from the above

This article elaborates the "doughnut framework" for societal and planetary boundaries. I am specifically asked to comment on the expert judgement component and on the manuscript more generally.

I am not an expert on indices but I do know that there is a large ecosystem of indices. For climate: Climate-Adapt of the EU

maintains 38 indices <https://climate-adapt.eea.europa.eu/en/knowledge/european-climate-data-explorer/overview-list> , explained in ETC-CCA Technical Paper 1/2020: Climate-related hazard indices for Europe. There is NOAA <https://psl.noaa.gov/data/climateindices/> , Climdex <https://www.climdex.org/learn/indices/> /limpact <https://climact-sci.org/indices/> World Meteorological Organization <https://community.wmo.int/en/climate-change-detection-and-indices> , IPCC https://www.ipcc.ch/report/ar6/wg1/downloads/report/IPCC_AR6_WGI_AnnexVI.pdf among many many others. For socio economic indices the list is longer. This manuscript presents an updated list of 35 social and ecological indices. These are presented without references to other lists and the unwary reader comes away with the notion that this particular list enjoys special status.

A fair amount of time is spent explaining the updates of previous versions of 2012, 2017. This followed a 2 step process (L 313) according to which the 2017 indicators were evaluated against criteria of Raworth, and second, using expert consultation. The expert methodology occupies the lowest tier of expert judgment methodology in science. For social indices 15 “Leading experts” (unidentified) were consulted. Nine experts responded and provided partially contradictory advice: (L 335) “ We gratefully received detailed comments and suggestions from nine reviewers (60% response rate). The reviewers provided a range of suggestions, at times contradictory, which we evaluated carefully to select the final set of dimensions, indicators, and descriptions included in the latest version of the Doughnut’s social foundation, as shown in Table 1. Although we invited leading experts with a wide range of expertise, we make no claims of a representative or exhaustive set of relevant insights, nor do we claim that all reviewers agree with all our choices – responsibility for the final indicator selection is ours.”

No further information is given as to what the disagreements were or how the authors made their choices. The scientific status of the results remains unclear.

For the ecological indices 10 experts were consulted of whom 6 responded.

(L 392) “The reviewers provided a range of suggestions, at times contradictory, which we evaluated carefully to select the final set of dimensions, indicators, and descriptions included in the latest version of the Doughnut’s ecological ceiling, as shown in Table 2. Although we have closely followed the planetary boundaries framework, we do not claim that the expert reviewers cover a representative or exhaustive set of relevant expertise, nor do we claim that all reviewers would agree with all the time-series indicators we have chosen. We take full responsibility for the final indicator selection.”

If this were a survey, which the authors say it is not, then 6 or 9 respondents is an insufficient number. If you invoke experts at all, then one should identify them, and give reasons why 40% didn’t respond. There should be a protocol saying how the experts were nominated, what precisely the experts were asked, what were the points of agreement and disagreement, and how disagreements were resolved in drawing their conclusions. The authors say that they are solely responsible for the conclusions. They sign off. Why then invoke experts at all? It seems more appropriate to say ‘we decided to do this based on our reading of the literature and discussions with colleagues’. Of course that doesn’t have much persuasive force.

Publication in NATURE may signal that the editors consider the selected indices better than all the others out there and that its method is worthy of emulation.

(Remarks on code availability)

Referee #6

(Remarks to the Author)

The paper presents an updated analysis of Doughnut Economics, integrating revised indicators to assess humanity’s progress with social and planetary boundaries between 2000 and 2021. The framework combines 35 indicators, covering social foundations and ecological ceilings, to provide a comprehensive view of whether humanity is meeting essential needs without exceeding environmental limits. The authors claim a concerning divergence: modest improvements in social indicators are outpaced by the worsening ecological conditions. The findings underscore that current efforts are insufficient, with planetary degradation progressing far more rapidly than the elimination of human deprivation, highlighting an urgent need for transformative action to achieve a just and sustainable future.

While the overall approach, i.e. juxtaposing the social and planetary boundaries together, is valuable I have a couple of major concerns to be addressed:

- The %-metrics between the domains are fundamentally not comparable due to fundamental differences in what they represent. For social boundaries, percentages denote the proportion of the global population experiencing a shortfall, such as lacking access to clean water or adequate healthcare, thus reflecting the extent of deprivation among people. In contrast, the percentage metrics for ecological boundaries represent the extent of overshoot beyond critical planetary limits, such as the safe level of CO2 concentration or nitrogen use. Hence, one of the main conclusion, comparing the speed of changes, is not justified, e.g. “our overarching finding is that ecological overshoot continues to worsen, and far more rapidly than social shortfall is improving. Social shortfall improves by 0–1% per year, based on the interquartile range of historical trends estimated” (line 172-174), but also the title and summary.
- While the social %-metrics are defined in a coherent way, the ecological metrics strongly depend on an assigned baseline that is differently defined for different boundaries. From a strict scientific statistical perspective %-metrics on variables that have not absolute zero should be avoided (e.g. we would not say 12°C is 20% warmer than 10°C), but I think it is acceptable

if one is aware of the implications. Here, one of the unwanted properties is for example, that with a lower base line the %-metric would be lower, e.g. as a thought experiment, if the preindustrial CO₂ would be, say only 250 ppm instead of 280 ppm the %-metric would not be 28.6 % but only 20% in 2000, although we would be further away from the baseline! This calls for at least a thorough sensitivity and uncertainty analysis

- While the authors do mention the importance of spatial variation, I believe it would now well be state-of-the-art and crucial to provide disaggregated analysis at least a couple of metrics. This will allow for a less broad-brush analysis and more nuanced statements. All the social metrics are available at least at country level and many planetary boundary metrics could be too, e.g. nutrient pollutions maps exist, CO₂ emissions by country etc. Including regional or country-level - at least: case - studies could help contextualize the findings and illustrate specific challenges and successes in navigating the Doughnut framework. Such examples could also provide insights into the differentiated roles of high-income versus low-income countries in meeting Doughnut goals.

- The paper provides limited actionable recommendations on how to accelerate progress. Future iterations should elaborate on specific, evidence-based recommendations, focusing on high-impact areas that could help achieve social goals while reducing ecological overshoot. This could also come from the regional analysis suggested above

- Please consider Mahecha et al. 2024 (<https://doi.org/10.5194/esd-15-1153-2024>) related to Figure 1. And good that the quantitative Figure 2 is not radial.

Minor comment:

- Figure S3 and S4 should be in the main text, but the captions are too similar for the fact that the %-metrics define something very different. Explain this in the caption.

(Remarks on code availability)

Version 2:

Reviewer comments:

Referee #1

(Remarks to the Author)

I want to thank the authors for their careful responses to my comments and questions. They have changed their article in line with most of my comments and explained why they opted not to in a few cases. I am satisfied with these changes and explanations and have no further comments.

(Remarks on code availability)

Referee #2

(Remarks to the Author)

Thank you for revising your manuscript and responding to my comments.

Regarding my first comment - essentially the "so-what" question, you respond that the updated doughnut serves basically as a monitor for progress. As for new directions you write "This calls for a deep renewal of both economic theory and practice, including by overcoming nations' structural dependency on GDP growth, so that they can instead reorient towards ecologically regenerative and socially distributive economic policies and outcomes." I understand, and I agree, but this is not so very concrete. What I was looking for in your paper is new an concrete insights that go beyond a monitoring function. In other words: I imagine I am sitting at my desk, looking at the doughnut progress report and asking myself, now what do I do? Although I agree, "a deep renewal of both economic theory and practice" doesn't really help me. And even if I set out on such a quest, how would the doughnut help me in very concrete ways, other than telling me it's not looking good on multiple fronts? That is still not clear to me.

This ties in with my second comment. If indeed that main contribution of your work is a monitoring function, then I believe a comparison with alternative frameworks is warranted. Coming back to UNEP's SCP-HAT, that framework's expressed purposes does not include research or policy directions, so it's OK that it doesn't provide such items. So, if you understand the doughnut as similarly and primarily providing a monitoring function, then I would be very interested in how this compared to SCP-HAT and others.

Regarding my comment about spillovers, I see now that you have included consumption-based indicators (thank you). What I could not find though is any mention of spillovers. You deal with shortfalls and overshoots, but some of those are strongly influenced by unequal trade relations, and outsourcings of the North to the South, which is surely an important trend, contributing to why we're actually seeing the global picture that you paint? I.e. social and environmental decline in the South driven by Northern affluence (see here the Scientists's warning by Wiedmann). So I was surprised that such outsourcing does not get a mention.

Other than that - as I said before, I like your work!

(Remarks on code availability)

Referee #3

(Remarks to the Author)

The Doughnut framework is a compelling way to communicate information about sustainable development that nicely includes both socioeconomic and ecological data. The revised version of the paper does a better job of making the main points in a more compelling manner than the original draft. The authors have done an adequate job of addressing most of my specific comments. The two exceptions are on my original comments numbers (4), (6) and (7). In comment (4), I said that the planetary boundary framework is questioned, especially for indicators that where the concerns are regional and not global in scale (N, P, HANPP). The authors did not address this point but merely said they did not have space to do an "in-depth critical assessment." I think it is a problem that deserves more attention when there are major flaws in an existing framework which the authors adopt without thinking critically about how it might affect results. I would like to see the authors think more deeply about their uncritical use of planetary boundaries. At this point, I do not insist that they revamp their analysis, but that they at least recognize major issues and discuss this briefly in the paper. Point (6) is a fairly specific point, which I still don't understand. Why is malnutrition falling while food insecurity is rising? I understand that these are different metrics and that this result is theoretically possible, but still it seems improbably in reality and I would like to see a clearer explanation of what is going on. Just saying it may happen doesn't really explain much for me. On point (7), I don't think it clear that all environmental problems tie to over-consumption and affluence. Specifically agricultural expansion, deforestation and other forms of habitat loss since 2000, which is the primary driver of biodiversity loss, has primarily occurred in low-income tropical countries, not in high-income temperate countries. Even middle-income countries, notably China, have transitioned to net reforestation. I am not convinced that the authors revisions in the paper address my comments nor did they do much to convince me or provide stronger evidence.

In the broad sweep of the paper, however, these are relatively minor points and while I would like to see the authors take a deeper look into these issues, I don't think this is enough to hold up the publication of a paper that I think many readers will find highly useful and will probably be widely cited.

(Remarks on code availability)

Referee #4

(Remarks to the Author)

The authors' responses and efforts to address several of the reviewer comments is greatly appreciated. At this stage, I have three specific comments/suggestions that can be addressed without too much of heavy-lifting.

1- The design of Figure 1 makes it look like the planetary boundaries for Ozone, air pollution, and Ocean acidification entirely safe, which is a bit misleading. PB 3.0 published last year indicates that two of them (air pollution, and Ocean acidification) are pretty close to being breached. Rockström, J., Donges, J.F., Fetzer, I., Martin, M.A., Wang-Erlandsson, L. and Richardson, K., 2024. Planetary Boundaries guide humanity's future on Earth. *Nature Reviews Earth & Environment*, 5(11), pp.773-788.

2- The author's response on the use of Palma ratio is quite intriguing -- Palma ratio, which measures the ratio of the share of GNI going top 10% to that going to bottom 40% income brackets, is an "economic" measure while "social cohesion" is a sociological variable.

3- The move to present a disaggregated analysis of three subgroups is helpful. Additionally, the authors should also acknowledge and flag for future research the critique that "the SJOS remains a very theoretical concept because a match does not exist between the Planetary Boundaries and the social components of the Doughnut." [Ferretto, A., Matthews, R., Brooker, R. and Smith, P., 2022. Planetary Boundaries and the Doughnut frameworks: A review of their local operability. *Anthropocene*, 39, p.100347].

(Remarks on code availability)

Referee #5

(Remarks to the Author)

no remarks to authors

(Remarks on code availability)

Referee #6

(Remarks to the Author)

The authors have addressed my comments mostly by discussing the limitations. The comment of a better disaggregation, which was also made by other reviewers, was addressed by a new analysis considering country income levels. However, this analysis was not about trends in time as the rest of the analysis, but rather on the broad spatial patterns.

As such, dividing countries into clusters (e.g., poorest 40%, middle 40%, richest 20%) and showing that richer nations have smaller social shortfalls but larger environmental footprints, while poorer nations have larger social deficits but smaller footprints, is not a new result. These findings echo a substantial body of research in human development and sustainability science. I am not deep into this (more economic) research, but e.g. references below come to mind.

Also, consumption-based assessments that include CO₂, water, nitrogen, and other resource footprints have been used by multiple researchers to demonstrate how high-income nations “overshoot” their fair shares of global environmental limits, whereas low-income nations remain within many planetary boundaries but fail to meet basic social needs. This aligns with previous studies that downscale planetary boundaries and compare them to observed consumption footprints at the national level.

Finally, the observation that wealthier groups—though much smaller in population—contribute a disproportionately large share of resource use or pollution, while poorer groups account for the majority of shortfalls in meeting social indicators, likewise reflects well-established patterns in the literature on environmental justice and global inequality. Together, these points confirm that while the data might be updated, and the specific breakdowns or indicators may differ, the broad conclusion is consistent with earlier scholarship.

So, now I tend to agree with the more critical reviewers of the last round, that the study is a nice science communication, but does not deliver substantially novel results of an original research article.

I haven't looked into the literature in depth, but it might be different if there are interesting trends also in the disaggregated data by income levels (But an economist will be better able to comment on this.). Actually, I do not understand why the authors do not report on trends at that level, while they do the global level.

To be constructive, to my mind it could be a good idea to cast this paper into a Review paper in Nature Reviews, certainly with a more comprehensive coverage of the past literature.

References:

O'Neill, D. W., Fanning, A. L., Lamb, W. F., & Steinberger, J. K. (2018). A good life for all within planetary boundaries. *Nature Sustainability*, 1(2), 88–95.

Wiedmann, T., Lenzen, M., Keyßer, L. T., & Steinberger, J. K. (2020). Scientists' warning on affluence. *Nature Communications*, 11(1), 3107.

Dorning, C., Hornborg, A., Abson, D. J., von Wehrden, H., Schaffartzik, A., Giljum, S., ... & Fischer-Kowalski, M. (2021). Global patterns of ecologically unequal exchange. *Ecological Economics*, 179, 106828.

Gidden, M. J., et al. (2019). Global emissions pathways under different socio-economic scenarios for use in CMIP6. *Geoscientific Model Development*, 12(4), 1443–1475.

In addition, United Nations Development Programme (UNDP) Human Development Reports, consistently show persistent deficits in health, education, and income in lower-income nations, in contrast to high-income countries.

(Remarks on code availability)

Doughnut of social and planetary boundaries monitors a world out of balance

Kate Raworth and Andrew L. Fanning

RESPONSES TO REFEREE COMMENTS

We are very grateful to the six Referees for their time and thoughtful comments on our study. We have made several significant changes to the text and figures, and we believe the manuscript is improved as a result. Please find below our detailed responses. The comments from the referees are in blue, while our responses to these are in black. All line numbers that we use refer to the revised manuscript.

Referee 1

This paper provides a first temporal assessment of whether the world is overshooting ecological boundaries and falling short of a social foundation. The paper does so with a carefully considered set of ecological and social dimensions and indicators from various data sources. The paper gives a clear picture of the world being on a dangerous path of increasingly overshooting ecological boundaries with comparatively small gains towards the social foundation. The paper is well written, the method is well explained, and the topic is certainly of relevance and interest to readers of Nature.

We are very grateful for the Reviewer's constructive review of our study.

I have three main comments and some minor points.

Main comments

1. Contribution: I think the contribution falls short of the bar of Nature, and recommend the authors add more value by breaking down the results at the country level.

Thank you, we have added a disaggregated analysis by country cluster (Lines 213-269).

I understand the contribution of the paper to be a revision of the doughnut framework and an application of the framework to temporal analysis. I find both impressive and relevant, yet in my understanding, this contribution falls short of the more groundbreaking contributions appearing in Nature (a call for the editor to make, to be sure). I certainly understand that revising the indicators is not as simple as it may sound, and one of the impressive aspects of this paper is to take something that is not simple at all and make it digestible for a general audience.

We are grateful for this frank assessment and appreciate the Reviewer's encouraging comments about our paper's digestibility for a general audience – this has indeed been one of our main aims, and we agree that it is not an easy thing to do.

Yet, I would like to see a greater contribution, possibly by extracting more relevant conclusions from the data. Most notably, it should be possible to say something at the country level about the social shortfall and ecological overshoot. Most of the indicators in the social foundation are already available at the country level, and even if some of these indicators have missing entries, those could be imputed. Less of the ecological indicators are available at the country level, yet the ones that are not could partially be computed at the country level using studies such as this one: <https://doi.org/10.1038/s41597-023-02041-1>

. Even if it is ultimately not possible to generate country level estimates for all indicators, it is possible for
sufficiently many to say something about which countries are responsible for overshooting the ecological
boundary, which countries have been able to make progress on the social foundation at the smallest
ecological cost etc.

We are grateful to the Reviewer for these comments, which are echoed by Reviewers 2, 3, 4, and 6, and
have inspired us to substantively revise the paper by adding a disaggregated analysis by country income
groups in the year 2017 (richest 20% of countries, middle 40% of countries, and poorest 40% of countries).
As the Reviewer notes, the social indicators allowed us to develop methods for the disaggregated analysis
that are fully comparable to the global results. For the ecological indicators, we collected global proxies
related to four of the planetary boundaries from a recent high-profile publication (also published in
Nature): <https://doi.org/10.1038/s41586-024-08154-w> . These new results indeed allow us to say
something about the countries that contribute the most to ecological overshoot and those that hold the
largest share of social shortfall (among other insights), which we find very important, and we believe this
increases the salience of our paper substantially.

2. Use of relative and subjective indicators: I recommend that the authors remove relative indicators and
strongly caveat subjective indicators. The indicators under the social foundation are partly subjective and
relative, both of which I find inappropriate for this exercise, and both of which are partially driving the
results. Most of the indicators with small or reverse progress in the social foundation fall under this
category.

Thank you for these recommendations, which we have implemented by replacing the relative indicators
and by providing further discussion of the three subjective indicators that we include in the Doughnut's
social foundation.

I think the relative indicators must be replaced, revised, or removed and the subjective indicators either the
same or heavily caveated. My understanding of the Voice and Accountability Index and the Control of
Corruption Index is that they rely on a relative scale. This means that even if all countries make substantive
progress, the global shortfall in these indicators need not fall. The authors should either reconstruct these
indicators such that they measure absolute progress or replace them altogether.

We are very grateful to the Reviewer for this insight, which really should have occurred to us, in hindsight.
We have replaced the Voice and Accountability Index and the Control of Corruption Index with non-relative
alternatives (Population living in countries governed by autocratic regimes (V-Dem Institute) and
Population stating that they perceive widespread corruption in government and business, respectively).
Notably, relative indicators were also included in the 2017 version of the Doughnut for these dimensions,
so we have added text describing why such relative indicators have been replaced, i.e. they are not
appropriate for our purposes of measuring performance with respect to absolute standards
(Supplementary Information Lines 456-460 and 506-509, respectively). Finally, given the Reviewer's
comments regarding subjective indicators, we are aware that our choice to replace the relative Control of
Corruption Index with a self-reported indicator tracking perceptions of corruption is less than ideal, but
after substantial exploration, we decided this was the best-available metric in terms of meeting our criteria
for indicator selection.

The subjective indicators (life satisfaction and food insecurity) can also be problematic. For example, the
Food Insecurity Experience Scale has been found to very weakly relate to objective measures of food
insecurity (<https://doi.org/10.1093/wber/lhad031>), casting doubt on what exactly it captures. Likewise, it
is well known that individuals (and cultures) interpret the life satisfaction scales differently and that any
conclusions from such measures may not be robust to alternative understanding of the scales (see here an

application to happiness, not life satisfaction:
<https://www.journals.uchicago.edu/doi/full/10.1086/701679?mobileUi=0>). To be sure, FIES and life
satisfaction also pick up relevant and important parts of a social foundation, but the measures could
certainly be caveated more by the authors.

Thank you for these comments and for sharing these helpful references. We note that the life satisfaction
indicator has been replaced with self-reported social support (as per the Reviewer's comment below), and
we have revised the descriptions of all self-reported metrics to mention the limitations of subjective
indicators in their respective indicator-specific sub-sections of Supplementary Information (Food Insecurity
Experience Scale: Lines 78-83; Social support: Lines 414-420; and Perceptions of corruption: Lines 498-500)

**3. Distinction between inequality and social cohesion: I recommend that the authors more clearly delineate**
**these two dimensions and rethink the indicators underpinning them.**

Thank you for this recommendation. After careful consideration, we have decided not to revise the social
cohesion and equality dimensions or their respective indicators. We agree that the previous descriptions
introducing these dimensions did not sufficiently distinguish their core difference, which is that social
cohesion focuses on the depth of inequalities within a society, while equality focuses on the breadth of
inequalities across a society. We have revised the descriptions to clarify this distinction (previously in
Methods, now in Supplementary Information Lines 19-26, due to space constraints).

**Both the inequality dimension and social cohesion dimension talk about inequality of opportunity.**
**Somewhat strangely, the only measure of inequality of outcomes (the Palma ratio) is not in the inequality**
**dimension but in the social cohesion dimension. The other indicator under social cohesion, life satisfaction,**
**surely is related to social cohesion, but life satisfaction is also related to all other dimensions tracked. I**
**don't think the authors sufficiently argue for its inclusion and I think the prior indicator used (self-reported**
**social support) seems to be closer to the concept the authors are after. Sure, it reduces the complexity of a**
**respondent's close relationship into a binary yes/no, yet the entire exercise in the social foundation is**
**about reducing complex relationships to binary states. My recommendation would be to move the Palma**
**Ratio to the equality dimension and replace life satisfaction with something more closely tied to social**
**cohesion.**

Thank you for these reflections and suggestions. As described in our previous response above, we have
envisioned equality and social cohesion as focusing on the breadth and depth of inequalities, respectively.
To reduce potential confusion, we have revised the social cohesion sub-section to remove mention of
inequality of opportunity. Based on these definitions, we believe that the Palma ratio is most coherently
placed within the social cohesion dimension so we have not moved it.

We had previously oscillated between selecting life satisfaction or social support as the most appropriate
metric, so we are grateful for the Reviewer's reflections here, which have convinced us to replace life
satisfaction with the self-reported social support metric used in the 2017 Doughnut. More broadly, we have
found it challenging to find concise and relatable dimension labels that sufficiently reflect their respective
domains, perhaps no more so than for social cohesion and equality.

**Smaller comments**

**The doughnut framework is a powerful concept for thinking about a social foundation and ecological**
**ceiling, yet as a visualization tool for presenting data, it has an important flaw if drawn the way in Figure 1.**
**The first sentence of the Wikipedia entry on Pie charts makes the point better than I can: "A pie chart (or a**
**circle chart) is a circular statistical graph which is divided into slices to illustrate *numerical proportion*" [I**
**added the asterisks]. As far as I understand, the areas in Figure 1 do not match numerical proportions,**

rather, one gets the visual impression that the overshoots are disproportionately larger than the
undershoots. To be sure, this is not a pie chart, but I do think people seeing this chart intuit that it
illustrates numerical proportions. This issue arises, I think, because the authors match the *height* of the
areas in Figure 1 with the shortfall/overshoot, and not the *areas* of Figure 1 with the shortfall/overshoot.
Solutions to this issue could be to present Figure 1 as Figure 2 or to mention in the figure note that the
areas are not proportional to actual shortfall/overshoot.

Thank you for this reflection, which is also raised by Reviewer 6. We have revised the figure caption of
Figure 1 and the new doughnut-shaped Figure 3 to clarify that values are proportional to the length of each
wedge, and we have also mentioned this point under Limitations (Lines 742-746). We have also visualised
Figure 2 and Figure 4 with bar charts to ensure readers can see comparable results from a non-radial
perspective.

In Table 1 and 2 where it says "% per year", should it be percentage points? Similarly, in discussions of
those two tables, I think "%" is often used instead of percentage points.

Thank you for flagging this point. This was an error on our part, and we have revised Tables 1 and 2 (and
relevant in-line text throughout) accordingly.

What explains the large jump in under-5 mortality around 2005 in Figure 3? It seems like something is off
with the data. I understand the authors want to be data takers, yet in this case, if some correction can be
made, this would increase the credibility of the findings. This large jump does not appear in the WDI:
<https://data.worldbank.org/indicator/SH.DYN.MORT?end=2022&start=1990&view=chart>

Thank you for this question. This indicator, like all social indicators in the Doughnut, is a measure of global
population in deprivation (not a global average rate, as in the WDI metric shared by the Reviewer). In this
case, it tracks the global population living in countries with under-five mortality rate exceeding 25 deaths
148 per 1,000 live births. As countries cross this minimum social standard, their populations are no longer
counted in the share of global shortfall, which can lead to visible 'jumps' when countries with large
populations cross the threshold, as China did in 2005 for this indicator.

There are five social indicators in our analysis with this 'Population living in countries...' formulation (under-
five mortality, lack of health services, income inequality/Palma, autocratic regimes, and homicide rate). We
have revised the Supplementary Information text describing each of the five indicators, noting their
sensitivity to countries with large populations. Although it would be preferable to aggregate within-country
shares of deprivation across all the social indicators, this is not currently possible for all indicators, given
existing data availability.

Under Acknowledgments, I think it would be appropriate to name the experts who provided feedback to
the draft set of dimensions and indicators.

Thank you, we have named all colleagues in the Acknowledgements section of the revised manuscript
(Lines 823-828).

The authors mention that a structural weakness of using broad-gauge purchasing power to monitor poverty
is that countries may have different prices for essential goods. Though certainly true in theory, in practice,
this concern does not appear to have much merit:
<https://www.aeaweb.org/articles?id=10.1257/app.3.2.137> .

Thank you for sharing this interesting reference. Given that the statement is accurate and it is beyond our
scope to enter further into the nuance of income poverty metrics, we have not revised these lines.

I appreciate the desire to track racial equality. My understanding is that in some countries, the type of
inequality the authors seek to measure is better described as ethnic inequality. Perhaps the indicator name
could be changed to reflect both types of inequalities (as indeed does the discussion in the supplementary
information).

Thank you for this encouraging comment and constructive reflection. We have revised the indicator
description in Table 1 to mention ethnic equality, and we have also mentioned ethnicity consistently
alongside race throughout the discussion in Supplementary Information (SI Lines 375-389)

The authors write that they “collected data on self-reported life satisfaction.” I doubt they actually
collected the data themselves, so perhaps the sentence could be rephrased to avoid confusion.

Thank you, we have revised accordingly.

Perhaps one sentence could be added explaining how human-induced radiative forcing complements the
CO2 concentration. What does the former capture that the latter does not? I am not doubting the choice,
only suggesting that one more sentence could be helpful for some reasons. Likewise, I recommend adding a
sentence on how 350 ppm CO2 relates to the Paris agreement and GMST increases.

Thank you for these suggestions. We have added two lines describing radiative forcing and the relation
between the climate boundaries and global temperature increases, respectively (Supplementary
Information Lines 553-558)

The authors mention that the indicators and dimensions used are related, and though this definitely will be
impossible (and probably also undesirable) to avoid, in at least one case, the indicators are not just related
but overlapping: Access to improved water and sanitation is part of the housing indicator and its own
indicator. What about removing this part from the housing indicator?

Thank you for this observation. We agree that it would be logical and ideal to avoid overlap in the housing
and water/sanitation indicators. However, there is no empirical basis to disaggregate the housing indicator,
as far as we are aware. We have revised the text to note this point (Supplementary Information Lines 344-
346).

**Referee 2**

This paper essentially reports on a data update of the Doughnut framework. It reports on an extension of
indicators, and expansion of the data foundation to a time series. The Doughnut framework has attracted
significant international attention, so its update is significant in itself.

Thank you for your time and consideration in reviewing our study.

**General comments:**

- I enjoyed reading the manuscript; my main comment is around the going forward section. Here, the first
paragraph is a summary of the preceding main text. The second paragraph talks about varying capacities
around the world. The third mentions “tools of doughnut economics”, but then settles on annual updates
to be provided. I would have hoped to read a bit more about the way forward. Now that we have
understood the dire situation, what do we do with your updated data? How – in concrete terms – can it
help us better in changing our current predicament? What tools exactly would you suggest that researchers
use in combination with your database, and how would they use them? What new and updated research
directions emerge from your new data? Some insights would be great.

Thank you for this comment. We have been inspired by the Reviewer’s invitation to share some thoughts
on the implications of the Doughnut results. This section has been completely revised and extended (now
entitled ‘Redefining and reorienting progress’), and we believe it is much improved (Lines 270-332).

- A comparison with similar time series data sets would be great. I’m sure there are a few, but I’d mention
one here, UNEP’s Sustainable Production and Consumption Hotspots Analysis Tool (SCP-HAT; [https://scp-](https://scp-hat.org)
[hat.org](https://scp-hat.org)) is one of them. Also here, you find time series, and many environmental, resource and social
indicators. Also this database is being regularly updated, under UNEP’s auspices. What can we learn from
the different frameworks? How does the Doughnut stand out?

Thank you for this suggestion to provide a comparison with other time-series databases. We certainly agree
that there is a fairly large and growing number of online tools and databases available that combine social
and ecological measures of performance. However, we believe it is beyond the scope of our paper to
conduct a comparative analysis with other databases in a manner that does justice to them, especially
given very limited space constraints. That said, we have added a discussion of conceptual distinctions
between the Doughnut and two complementary frameworks (decent living standards and safe and just
Earth-system boundaries), which we think captures some of the spirit of the Reviewer’s suggestion (Lines
314-326). Perhaps most notably, the Reviewer’s useful question ‘How does the Doughnut stand out?’
inspired us to outline contributions of the Doughnut framework to shifting the conception, metrics, and
directionality of progress in the 21st century, which we believe are important to articulate (Lines 271-283).

- I mention SCP-HAT, because it includes spill-overs and outsourcing, which I believe the Doughnut does
not. This is a major factor in monitoring SDGs. Affluent countries outsource dirty and unfair production to
regions with lax regulations, mostly in the global south. This outsourcing contributes majorly to countries’ –
and the world’s – performance on many indicators. See for example a recent publication on polarising
trends on multiple indicators¹, which I believe speaks very well to your observations, but adds some
analytical rigour. Outsourcing effects are also explicitly reflected for example in SDGs 8 and 12, through the
material footprint indicator², which is monitored by the SCP-HAT website.

Thank you for this comment. The global Doughnut monitors the performance of humanity as a whole, so it
does not include spill-overs or outsourcing between countries, as the Reviewer notes. However, the
importance of accounting for such inequalities is also noted by Reviewers 1, 3, 4, and 6, and have inspired
234 us to disaggregate the global Doughnut based on a country-cluster analysis of the poorest 40% of countries,
middle 40% of countries, and richest 20% of countries (Lines 213-269). Of particular relevance to the
Reviewer’s comment, we incorporate six downscaled ecological indicators, which are all consumption-
based environmental footprint metrics that account for spillovers and outsourcing via international trade.

Specific comments:

- This is a minor comment. You write: “static snapshot” and “dynamic monitor” when you refer to your time
series. In my understanding, the term “dynamic” implies some sort of intertemporal interaction. What you
have is a series of static snapshots, but that does not make the time series “dynamic”, as in the sense that a
previous year acts on the following year by some sort of acceleration or retarding. I realise this might be
word splitting, but I thought I mention it nevertheless. A truly multi-indicator dynamic analysis on the SDGs
was recently published, using CGE analysis.³

Thank you for this observation and for sharing this recent publication. We agree with the point, we have
removed the word ‘dynamic’, and we are grateful for the improved accuracy (not word splitting!).

- You state that “we acknowledge the interdependency of many indicators. While we do not analyse such
complex interdependencies formally here...” because the radar diagram “conveys a visual sense of holistic

interconnection". However, one could argue that in portraying (visually) a suite of indicators without
consideration of interconnection might provide a distorted message. Imagine I chose to portray
environmental and resource issues choosing i) GHG emissions, ii) energy use, iii) deforestation, iv)
biodiversity loss and v) ozone layer depletion. Indicators i) and ii), as well as i) and iii), and iii) and iv) are
highly correlated, and would yield the same trends, whereas v) is much less correlated to i)-iv). Portraying
those would give the sub-suite i)-iv) undue weight. There are methods that deal with criteria overlap, such
as the Choquet integral⁴⁻⁶. You might want to consider this, eg for the radar diagram, or in case you ever
want to establish country rankings.

Thank you for this comment and example. We agree that interconnections are important and an active field
of research but we believe that it is beyond the scope of this study to develop and apply methods for them
here.

Having said this, overall this is a great effort, and will be used and cited a lot.

We appreciate this encouraging comment, and we thank the Reviewer for contributing their time and
expertise.

References

1 Malik, A. et al. Polarizing and equalizing trends in international trade and Sustainable Development Goals.
Nature Sustainability, doi:10.1038/s41893-024-01397-5 (2024).

2 Lenzen, M. et al. Implementing the material footprint to measure progress towards Sustainable
Development Goals 8 and 12. Nature Sustainability 5, 157-166, doi:10.1038/s41893-021-00811-6 (2022).

3 Han, S. et al. Prospects for global sustainable development through integrating the environmental
impacts of economic activities. Nature Communications 15, 8424, doi:10.1038/s41467-024-52854-w (2024).

4 Marichal, J.-L. An axiomatic approach of the discrete Choquet integral as a tool to aggregate interacting
criteria. IEEE Transactions on Fuzzy Systems 8, 800-807 (2000).

5 Rowley, H. V., Geschke, A. & Lenzen, M. A practical approach for estimating weights of interacting criteria
from profile sets. Fuzzy Sets and Systems 272, 70-88, doi:http://dx.doi.org/10.1016/j.fss.2015.01.011
(2015).

6 Marichal, J.-L. & Roubens, M. Determination of weights of interacting criteria from a reference set.
European Journal of Operational Research 124, 641-650 (2000).

Referee 3

Doughnut economics is a wonderful communication tool to show progress towards or away from
sustainable development goals. It frames meeting minimum socio-economic goals while not exceeding
environmental limits in an intuitive and compelling manner. This paper updates socio-economic and
environmental indicators and offers a few modest extensions of the prior published work on doughnut
economics.

We are grateful to the Reviewer for making time to review our study.

The main message of the paper is that global society is seriously off track ("deeply out of balance") in
meeting sustainable development objectives. The paper finds that there have been minor gains in many

socio-economic goals but a rapid deterioration in Earth's life-support systems over the past two decades.
These results are consistent with the findings of other recent publications, including the UN Sustainable
Development Report, Global Environment Outlook, World Development Report, IPCC AR6, the IPBES Global
Assessment, Richardson et al. (2023), Rockström et al. (2009), Steffen et al. (2015), and others. Much of the
paper is indeed built from data sources developed by these other assessments (e.g., sustainable
development indicators contained in the Sustainable Development Report, and planetary boundary
indicators contained in Richardson et al. 2023 and earlier planetary boundary papers). I don't think the
paper offers much that is of interest in addition to what is in these assessments, and I did not learn much
that is new from reading the paper. Given this, I don't think the paper should be published in Nature.

We appreciate the Reviewer's frank assessment. We believe the revised manuscript offers substantially
improved and novel findings, notably by including a disaggregated country-cluster analysis in a cross-scale
comparable framework (Lines 213-269). We also articulate contributions that the Doughnut framework
makes to redefining and reorienting progress in the 21st century (Lines 270-332), which we believe are
useful and unique distinctions compared to the other frameworks mentioned by the Reviewer (although we
do indeed build upon them).

The only argument at present I can see for publishing the paper in Nature is that it would provide more of a
spotlight on what is a very useful communication tool, one that hasn't previously been published in a high-
profile scientific journal like Nature.

Thank you for this comment. We believe the revised manuscript addresses the novelty concerns raised by
the Reviewer regarding our Analysis.

Other than this argument, I think the authors would need to revamp the paper in a significant way to
provide notable contributions to what already exists in the literature to make the case for publication.

Thank you for this comment, which we believe is consistent with points made by Reviewer 1, 2, 4 and 6. We
have been inspired to substantively revise the study based on the Reviewer comments, notably by adding a
disaggregated analysis by country income clusters, which concisely reveals considerable inequalities in
social shortfall and ecological overshoot within the global doughnut (Lines 213-216). In our view, these new
results, and the implications they help make visible, provide a notable and useful contribution from our
study to the literature.

I have a number of relatively specific and mostly minor comments that could be addressed in a revision
should the editorial decision be to proceed (or to help the authors in submitting to another journal).

Thank you for sharing these helpful comments.

1. The opening sentence could be rewritten to be clearer. The way it is written it seems like you are
claiming that there was an assumption of endless growth in GDP, which would be hard to demonstrate,
rather than simply stating that growth in GDP was taken to be the predominant measure of success, which
could be more easily substantiated.

Thank you, we have revised the opening sentence accordingly (Lines 40-42).

2. While I think it is accurate to say that growth in GDP was a dominant measure of success since at least
the 1950s when measurement of GDP became widespread, there have been counterarguments present
from the beginning. Even the economists who pioneered the measurement of GDP knew that GDP was NOT
a measure of welfare but simply a measure of income as measured by flows of values in market economies.

Thank you for this comment, which touches on a fascinating history of considerable interest to us.
However, for the purposes of this paper, we have not included additional nuance on this point due to a
very limited word-count, and given the Reviewer's agreement on the accuracy of the statement.

I also think it is too optimistic to say that society as a whole has outgrown its infatuation with growth in
GDP as a measure of success (just look at the rhetoric in almost any national election).

Thank you for this reflection. We have replaced 'The emerging 21st century conception...' with 'An
emerging 21st century conception...' – a light revision, to be sure, but we believe it reduces the implied
optimism substantially (Lines 42-43).

3. Lines 55-62: this paragraph, which is mostly self-promotional, could be deleted.

Thank you for this suggestion. We have revised this paragraph in any case as part of broader revisions to
the Introduction. That said, we believe that it is worthwhile to include a line for readers to have a sense of
the Doughnut's uptake and influence beyond the scientific community (Lines 67-70).

4. The paper relies on the "planetary boundaries" framework for characterizing ecological overshoot. Like
doughnut economics, the planetary boundaries framework is a wonderful for simple communication and is
widely referenced. However, the planetary boundaries framework has also been criticized as lacking sound
scientific foundations. For example, unlike greenhouse gases that mix in the global atmosphere, excess
nitrogen and phosphorus cause problems in specific regions. It doesn't make logical sense to talk about
nitrogen and phosphorus as having a global boundary, even if one derives it by taking averages of regional
levels. I think it would be much more informative to talk about the share of regions that exceed critical
levels, similar to what is done on the socio-economic side where it is the share of people who live below
certain standards. I am also not convinced that HANPP is a meaningful measure of a planetary boundary. It
is a measure of the scale of human activity, but does it really indicate an ecological overshoot in the way
that loss of biodiversity or increases in greenhouse gas emissions, N and P overloads, or other metrics do?
Having some critical assessment of planetary boundaries would add value to the work. Simply presenting
the work of Richardson et al. (2023) again does not add much value.

Thank you for these comments and suggestions. While we agree that it is important to note that the
planetary boundaries concept has been criticised, we think that it is out of scope of the present paper to
provide an in-depth critical assessment. We have revised the Introduction to add this point (Lines 82-85)

5. On the socio-economic side, the paper uses many of the sustainable development indicators. There are
many more sustainable development indicators that are not used. A stronger justification for why you use
this set of indicators and not others would strengthen the paper. Some thresholds appear to be fairly
arbitrary (e.g., 5 homicides per 100,000 per year). When looking at trends, arbitrary cutoff values are less
important, and one could do further analysis to show that the trends are robust to other reasonable
(arbitrary) choices of the cutoff value.

We are grateful for this comment. We have revised the text to acknowledge that the set of social indicators
we have selected is open to debate, noting that in practice we have found the six criteria we followed to be
quite limiting (Lines 563-566), and we also acknowledge that some indicators lack officially recognised
thresholds (such as the homicide rate; Lines 763-766). That said, we have conducted an internal review of
this set of indicators with leading experts, and we provide more than 7,000 words of indicator-specific
discussion relating to the set of social indicators in Supplementary Information. We believe these methods
provide sufficiently strong justification for our indicator selection, without denying that they are indeed
open to contestation.

6. It is puzzling to me that measures of malnutrition show declining malnutrition while measures of food
insecurity are rising. Is there an explanation?

Thank you for this question. We have revised the food indicator descriptions to discuss differences between
them, limitations, and a possible explanation for their diverging trends (Supplementary Information Lines
51-58). In essence, although the indicators are not strictly comparable, we believe that the main difference
is that our inclusion of people with 'moderate to severe' food insecurity is a higher minimum standard than
the one used to define undernourishment (which has been shown to be strongly correlated with 'severe'
food insecurity in this article: <https://ourworldindata.org/food-insecurity>).

7. In lines 230-231, you state that ecological overshoot "is overwhelmingly being driven by excessive
consumption and control over production processes by the affluent." I think this statement is factually
correct for greenhouse gas emissions and several other metrics, but is not clearly true for some others,
such as loss of habitat. In recent decades agricultural expansion is happening in low-income countries,
driven by rising population and stagnant yields, but not in high-income countries. A close look at the data
also shows that rising land use in low-income countries is primarily driven by rising domestic consumption
rather than rising net exports of crops to high-income countries.

Thank you for these comments. Although we are not entirely clear which specific data and results the
Reviewer is referring to, we believe that this claim is now better-substantiated in our revised study by the
respective levels of the six consumption-based environmental footprint indicators included in our country-
cluster analysis of per capita overshoots, as shown in Figure 4. Perhaps especially by the species-loss
footprint and the HANPP footprint, given the Reviewer's example on habitat loss, which both show
increasing levels of per capita overshoot at higher levels of income. As such, we have not removed the spirit
of the statement quoted by the Reviewer (which we believe is also substantiated by the studies we have
referenced), although we note that this specific sentence has been revised as part of broader changes to
the Introduction (Lines 90-93)

8. I did not find the exercise of simply projecting past trends into the future to be very informative.
However, if you are going to do this, then I would also indicate that the ocean acidification indicator,
though currently within its boundary, will cross its boundary by 2050 unless the trend is changed.

Thank you for this comment. Although we regret that the Reviewer did not find the trend analysis
informative, we fear that they may have misunderstood our approach, as we do not project past trends
into the future. We estimated historical trends for each social and ecological indicator, and we compared
these to the rates of change required to eliminate 2022 levels of social shortfall and ecological overshoot by
2030 and 2050, respectively. Although we agree this is a simple method, we believe it provides an intuitive
comparison of the current trend in relation to what is required.

In terms of the likely year that the ocean acidification boundary, a simple linear projection of our results
suggest that it will likely be crossed this year - 2025. However, our study does not project past trends into
the future, so we have noted the very small share of safe space remaining is closing quickly (Lines 162-165).

9. The final paragraph (lines 237-244) seems more like a sales pitch for the lab than a concluding paragraph.
I would largely delete the current paragraph, keeping only the parts about the importance of data for
bending the curves on trendlines. It would also be helpful to indicate awareness of the difficulties of
overcoming barriers to change needed for successfully bending the curves.

Thank you for this comment. We have fully revised and extended the Discussion section, including
removing reference to Doughnut Economics Action Lab in the final paragraph, and also discussing barriers,

notably the need to overcome structural dependency on GDP growth in order to reorient towards
ecologically regenerative and socially distributive economic policies and outcomes (Lines 270-332)

**Referee 4**

"The Doughnut" has made a useful contribution to the public debates on social and ecological planetary
boundaries. Much of the past scholarship on the Doughnut has eschewed critical reflections about the
limitations and ways to address them. This manuscript seems to acknowledge these limitations by hand-
waiving at questions of colonialism and inequalities. Yet, the manuscript makes no advances in offering an
analytical solution to these gaps and critiques. This would be a missed opportunity.

Thank you for making time to review our study. Our primary response to the Reviewer's invitation to offer
an analytical solution is to include a country-cluster analysis based on income percentile groups (poorest
40% of countries, middle 40% of countries, and richest 20% of countries). We believe this disaggregated
analysis, including the cross-scale comparable framework that we have developed to hold the results
coherently alongside our global analysis, makes a substantive contribution to making visible inequalities
masked by global aggregates (Lines 213-269)

1) In the spirit of doing good science, I encourage the authors to look for and engage with some critical
arguments. More importantly, tackling some of these critiques also offer an opportunity to advance
debates on transformational change in a world-ridden by multiple and interconnected crises of wealth
inequalities, environmental degradation, and the climate crisis. The following may be a good starting point
but **authors should undertake a more systematic review of critical takes on the Doughnut**: Drees, L.,
Luetkemeier, R. and Kerber, H., 2021. Necessary or oversimplification? On the strengths and limitations of
current assessments to integrate social dimensions in planetary boundaries. *Ecological indicators*, 129,
p.108009.

Thank you for this invitation and for sharing this publication. We have included a sentence acknowledging
critical assessments of the Doughnut framework, with reference to this study and others (Lines 85-87).

2) By now there is abundant evidence to suggest that global inequalities and ecological degradation are
deeply interconnected. Yet, the Doughnut continues to perpetuate this binary between the social
foundations and "the humanity's" footprint. While the authors recognize this, they should also draw out
implications, and preferably, offer a preliminary empirical analysis of the extent to which global inequalities
are responsible for the massive overshoot on ecological indicators.

Thank you for this comment, which aligns with points made by Reviewers 1, 2, 3, and 6. Of particular
relevance to this point, our disaggregated analysis allows us to show that the richest 20% of countries are
home to just 15% of the world's population, yet they collectively contribute more than 40% to global
ecological overshoot across a majority of per capita boundaries (Lines 259-269).

This is especially relevant given the evidence that the burden of addressing the environmental crisis has
been imposed on countries with high levels of economic inequalities and missing or dysfunctional
democracies. Kashwan, P., 2017. Inequality, democracy, and the environment: A cross-national analysis.
*Ecological Economics*, 131, pp.139-151.

Thank you for this comment and for sharing this interesting publication, which we have referenced in the
Introduction (Ref. 36).

3) The IEA data suggesting that 9% of global population lacks access to electricity has proven to be
unreliable. See, Min, B., O'Keeffe, Z.P., Abidoye, B., Gaba, K.M., Monroe, T., Stewart, B.P., Baugh, K. and
Nuño, B.S.A., 2024. Lost in the dark: A survey of energy poverty from space. *Joule*.

Thank you for sharing this comment and for sharing this interesting publication. We are aware of the
limitations of the access to electricity indicator, and that interesting alternatives are being demonstrated by
researchers - including this study by Min et al., which we have referenced - but alternative time-series
metrics are not yet available globally as far as we are aware (Supplementary Information Lines 282-284)

4) The authors might do well to reflect deeply on the extent to which global modeling of these scenarios
obscures multi-level complexities and promotes copy-cat aggregate analyses in contexts that differ wildly.
Schultz, B., Brockington, D., Coleman, E.A., Djenontin, I., Fischer, H.W., Fleischman, F., Kashwan, P.,
Marquardt, K., Pfeifer, M., Pritchard, R. and Ramprasad, V., 2022. Recognizing the equity implications of
restoration priority maps. *Environmental research letters*, 17(11), p.114019.

Thank you for this suggestion. We have referenced this interesting study in a revised and extended version
of the Discussion (Lines 292-295).

5) This is relevant for Global South countries that lack a rule of law framework and quality datasets but also
in the Global North countries with extreme inequality: Tønnessen, M. Wasted GDP in the USA. *Humanit Soc
Sci Commun* 10, 681 (2023). <https://doi.org/10.1057/s41599-023-02210-y>

Thank you for this comment. We have revised the text to mention the desirability of cross-country
comparable data that would allow such within-country inequalities to be made visible in the Doughnut
(Lines 301-302 and 755-758), but it is not currently possible with available data, especially for the social
indicators.

Referee 5

A updated doughnut graph of ecological and social indices

B these indices are not positioned in the large space of climate and social indices. Originality not clear. Only
the doughnut presentation seems original, which is science communication rather than original science.

C expert judgment method is under par, experts names and qualifications, method of recruitment,
rationales etc not given

D no treatment of uncertainties

F retrievable from the above

This article elaborates the “doughnut framework” for societal and planetary boundaries. **I am specifically**
**asked to comment on the expert judgement component** and on the manuscript more generally.

Thank you for making time to review our study. It seems to us that points A, B, C, D, E, and F above are
notes, rather than comments or suggestions that we are able to address. In terms of the specific focus on
the ‘expert judgement’ component, our main response is that we agree with the Reviewer’s assessment
below. We have reframed this part of our methodological process wholly as an internal review with
colleagues to avoid association with formal expert elicitation methods. We have realised that it was an
error on our part to frame these methods as an expert consultation in the previous version, and we are
grateful for the Reviewer’s time and expertise.

I am not an expert on indices but I do know that there is a large ecosystem of indices. For climate: Climate-
Adapt of the EU maintains 38 indices [https://climate-adapt.eea.europa.eu/en/knowledge/european-](https://climate-adapt.eea.europa.eu/en/knowledge/european-climate-data-explorer/overview-list)
[climate-data-explorer/overview-list](https://climate-adapt.eea.europa.eu/en/knowledge/european-climate-data-explorer/overview-list) , explained in ETC-CCA Technical Paper 1/2020: Climate-related hazard
indices for Europe. There is NOAA <https://psl.noaa.gov/data/climateindices/> , Climdex
<https://www.climdex.org/learn/indices/> limpact <https://climpact-sci.org/indices/> World Meteorological

Organization <https://community.wmo.int/en/climate-change-detection-and-indices> , IPCC
https://www.ipcc.ch/report/ar6/wg1/downloads/report/IPCC_AR6_WGI_AnnexVI.pdf among many many
others. For socio economic indices the list is longer. This manuscript presents an updated list of 35 social
and ecological indices. These are presented without references to other lists and the unwary reader comes
away with the notion that this particular list enjoys special status.

Thank you for this comment. It is beyond the scope of our study to create a detailed compendium of the
many indicators, indices, and databases that track one or more features related to ecological and social
performance. Our view is that there is a shared understanding by readers of Nature that many indicators
and indices exist, and we believe that we have sufficiently documented our choices and indicator-specific
data sources in Methods and especially Supplementary Information.

A fair amount of time is spent explaining the updates of previous versions of 2012, 2017. This followed a 2
step process (L 313) according to which the 2017 indicators were evaluated against criteria of Raworth, and
second, using expert consultation. The expert methodology occupies the lowest tier of expert judgment
methodology in science. For social indices 15 “Leading experts” (unidentified) were consulted. Nine experts
responded and provided partially contradictory advice:

(L 335) “ We gratefully received detailed comments and suggestions from nine reviewers (60% response
rate). The reviewers provided a range of suggestions, at times contradictory, which we evaluated carefully
to select the final set of dimensions, indicators, and descriptions included in the latest version of the
Doughnut’s social foundation, as shown in Table 1. Although we invited leading experts with a wide range
of expertise, we make no claims of a representative or exhaustive set of relevant insights, nor do we claim
that all reviewers agree with all our choices – responsibility for the final indicator selection is ours.”

No further information is given as to what the disagreements were or how the authors made their choices.
The scientific status of the results remains unclear.

Thank you for this assessment. We have substantially revised this sub-section by moving the comparison of
2012 and 2017 updates to Supplementary Information, and, importantly for the Reviewer’s point above, by
reframing the ‘expert consultation’ as an ‘internal review with colleagues’ (Lines 557-563). We have also
revised the Acknowledgements section by naming all colleagues who have contributed to data collection
and internal review of the social indicators (Lines 823-828).

For the ecological indices 10 experts were consulted of whom 6 responded.

(L 392) “The reviewers provided a range of suggestions, at times contradictory, which we evaluated
carefully to select the final set of dimensions, indicators, and descriptions included in the latest version of
the Doughnut’s ecological ceiling, as shown in Table 2. Although we have closely followed the planetary
boundaries framework, we do not claim that the expert reviewers cover a representative or exhaustive set
of relevant expertise, nor do we claim that all reviewers would agree with all the time-series indicators we
have chosen. We take full responsibility for the final indicator selection.”

Thank you. Similar to the social foundation section, we have substantially revised this sub-section by
moving the comparison of 2012 and 2017 updates to Supplementary Information, and, importantly for the
Reviewer’s point above, by reframing the ‘expert consultation’ as an ‘internal review with colleagues’ (Lines
586-592). We have also revised the Acknowledgements section by naming all colleagues who have
contributed to data collection and internal review of the ecological indicators (Lines 823-828).

If this were a survey, which the authors say it is not, then 6 or 9 respondents is an insufficient number. If
you invoke experts at all, then one should identify them, and give reasons why 40% didn’t respond. There
should be a protocol saying how the experts were nominated, what precisely the experts were asked, what

were the points of agreement and disagreement, and how disagreements were resolved in drawing their
conclusions. The authors say that they are solely responsible for the conclusions. They sign off. Why then
invoke experts at all? It seems more appropriate to say 'we decided to do this based on our reading of the
literature and discussions with colleagues'. Of course that doesn't have much persuasive force.

Thank you for these comments and reflections. Again, we agree that the methods we have employed are
more akin to an internal review with colleagues, and we have reframed accordingly.

Publication in NATURE may signal that the editors consider the selected indices better than all the others
out there and that its method is worthy of emulation.

We are grateful for the Reviewer's time and expertise. Although we agree the framing of an internal review
with colleagues more accurately reflects this single aspect of our methodological approach, we want to
underscore that it was nevertheless a deeply enriching process and we are deeply grateful to colleagues for
their contributions. In addition, we believe that other features of our methodological approach in the
revised manuscript (i.e. data collection and documentation, analysis and visualisation, results and
implications) are substantially improved and useful contributions to knowledge in their own rights, with
thanks to the Reviewers' comments.

**Referee 6**

The paper presents an updated analysis of Doughnut Economics, integrating revised indicators to assess
humanity's progress with social and planetary boundaries between 2000 and 2021. The framework
combines 35 indicators, covering social foundations and ecological ceilings, to provide a comprehensive
view of whether humanity is meeting essential needs without exceeding environmental limits. The authors
claim a concerning divergence: modest improvements in social indicators are outpaced by the worsening
ecological conditions. The findings underscore that current efforts are insufficient, with planetary
degradation progressing far more rapidly than the elimination of human deprivation, highlighting an urgent
need for transformative action to achieve a just and sustainable future.

Thank you for making time for this constructive review of our study.

While the overall approach, i.e. juxtaposing the social and planetary boundaries together, is valuable I have
a couple of major concerns to be addressed:

• The %-metrics between the domains are fundamentally not comparable due to fundamental differences
in what they represent. For social boundaries, percentages denote the proportion of the global population
experiencing a shortfall, such as lacking access to clean water or adequate healthcare, thus reflecting the
extent of deprivation among people. In contrast, the percentage metrics for ecological boundaries
represent the extent of overshoot beyond critical planetary limits, such as the safe level of CO2
concentration or nitrogen use. Hence, one of the main conclusion, comparing the speed of changes, is not
justified, e.g. "our overarching finding is that ecological overshoot continues to worsen, and far more
rapidly than social shortfall is improving. Social shortfall improves by 0–1% per year, based on the
interquartile range of historical trends estimated" (line 172-174), but also the title and summary.

Thank you for this very useful comment. We agree with the Reviewer and are grateful for the clear
expression of an uneasy concern that we had also felt with the previous comparisons between the social
and ecological domains (but lacked the clarity to articulate). We have removed such direct comparisons
across the social and ecological domains throughout, and noted this point (Lines 640-644).

• While the social %-metrics are defined in a coherent way, the ecological metrics strongly depend on an
assigned baseline that is differently defined for different boundaries. From a strict scientific statistical

perspective %-metrics on variables that have not absolute zero should be avoided (e.g. we would not say
12°C is 20% warmer than 10°C), but I think it is acceptable if one is aware of the implications. Here, one of
the unwanted properties is for example, that with a lower base line the %-metric would be lower, e.g. as a
thought experiment, if the preindustrial CO2 would be, say only 250 ppm instead of 280 ppm the %-metric
would not be 28.6 % but only 20% in 2000, although we would be further away from the baseline! This calls
for at least a thorough sensitivity and uncertainty analysis

Thank you for this illuminating example. With respect to the preindustrial baselines used for the ecological
indicators, we have taken these directly from the planetary boundaries framework. Although we agree that
the overshoot values are sensitive to these baselines, it is beyond the scope of our study to assess the
implications of different preindustrial baselines on the planetary boundaries. We have added some words
noting the dependence on preindustrial baselines (Lines 644-646).

• While the authors do mention the importance of spatial variation, I believe it would now well be state-of-
the-art and crucial to provide disaggregated analysis at least a couple of metrics. This will allow for a less
broad-brush analysis and more nuanced statements. All the social metrics are available at least at country
level and many planetary boundary metrics could be too, e.g. nutrient pollutions maps exist, CO2 emissions
by country etc. Including regional or country-level - at least: case - studies could help contextualize the
findings and illustrate specific challenges and successes in navigating the Doughnut framework. Such
examples could also provide insights into the differentiated roles of high-income versus low-income
countries in meeting Doughnut goals.

Thank you for this comment, which is aligned with similar points by Reviewers 1, 2, 3, and 4. As noted in
those comments, we have added a disaggregated analysis by country income clusters (poorest 40% of
countries, middle 40% of countries, and richest 20% of countries) that we believe indeed adds useful
insights into the differentiated roles of high-income versus low-income countries (Lines 213-269).

• The paper provides limited actionable recommendations on how to accelerate progress. Future iterations
should elaborate on specific, evidence-based recommendations, focusing on high-impact areas that could
help achieve social goals while reducing ecological overshoot. This could also come from the regional
analysis suggested above

Thank you for this suggestion. We have fully revised and extended the Discussion section, indeed by
drawing on the country-cluster analysis, and also including a broader call to overcome a structural
dependence on GDP growth in economic policymaking, among other points (Lines 270-332)

• Please consider Mahecha et al. 2024 (<https://doi.org/10.5194/esd-15-1153-2024>) related to Figure 1.
And good that the quantitative Figure 2 is not radial.

Thank you for sharing this publication. We have noted the limitations of radial representations of indicators
in data visualisation, with reference to this study (Lines 744-748).

**Minor comment:**

• Figure S3 and S4 should be in the main text, but the captions are too similar for the fact that the %-
metrics define something very different. Explain this in the caption.

We are glad that the Reviewer finds these figures informative, and we appreciate this suggestion but we
cannot include these charts in the Main text given the limited space for display items, especially now that
we have included the country-cluster analysis in this revised version. We have revised relevant figure
captions throughout to clarify the ranges of social shortfall and ecological overshoot.

Doughnut of social and planetary boundaries monitors a world out of balance

Andrew L. Fanning and Kate Raworth

RESPONSES TO REFEREE COMMENTS

Thank you to the six Referees for making time and sharing thoughtful comments on our revised study. We have made several significant changes to the text, which we believe improve the manuscript further. Please find below our detailed responses. The comments from the Referees are in blue, while our responses to these are in black. All line numbers that we use refer to the most recent revised manuscript.

Referee 1

I want to thank the authors for their careful responses to my comments and questions. They have changed their article in line with most of my comments and explained why they opted not to in a few cases. I am satisfied with these changes and explanations and have no further comments.

We are very grateful for the Reviewer's time and expertise throughout the review process of our study.

Referee 2

Thank you for revising your manuscript and responding to my comments.

Regarding my first comment - essentially the "so-what" question, you respond that the updated doughnut serves basically as a monitor for progress. As for new directions you write "This calls for a deep renewal of both economic theory and practice, including by overcoming nations' structural dependency on GDP growth, so that they can instead reorient towards ecologically regenerative and socially distributive economic policies and outcomes."

I understand, and I agree, but this is not so very concrete. What I was looking for in your paper is new and concrete insights that go beyond a monitoring function. In other words: I imagine I am sitting at my desk, looking at the doughnut progress report and asking myself, now what do I do? Although I agree, "a deep renewal of both economic theory and practice" doesn't really help me. And even if I set out on such a quest, how would the doughnut help me in very concrete ways, other than telling me it's not looking good on multiple fronts? That is still not clear to me.

Thank you for this comment, which touches on an important area for us. However, after considerable reflection, we have concluded that incorporating meaningful concrete actions/strategic guidance in a manner that does justice to the diversity of distinct contexts simply cannot be feasibly articulated within the scope and space constraints of the present study. That said, we have revised the final paragraph substantively in this direction by stating our aim for the Doughnut results to be used as a tool to inform action-led initiatives, with specific reference to our experience through Doughnut Economics Action Lab (Lines 346-357).

This ties in with my second comment. If indeed that main contribution of your work is a monitoring function, then I believe a comparison with alternative frameworks is warranted. Coming back to UNEP's SCP-HAT, that framework's expressed purposes does not include research or policy directions, so it's OK

that it doesn't provide such items. So, if you understand the doughnut as similarly and primarily providing a
monitoring function, then I would be very interested in how this compared to SCP-HAT and others.

We are grateful for this reflection, which we have considered carefully. However, we have not conducted a
formal comparative analysis of the Doughnut with alternative monitoring frameworks as part of the
present study due to very limited space constraints, as stated in the previous round of review. In our view,
such an analysis would be a non-trivial task to design and carry out with care, and we believe the result, if
done well, would be an informative contribution that deserves far more attention than we can provide here
(i.e. one or two sentences in the Main text, and the rest buried in Supplementary Information).

We believe UNEP's SCP-HAT database referenced by the Referee can provide a useful illustration of
methodological questions that arise when designing comprehensive selection criteria for a comparative
analysis of alternative frameworks. There is no question to us that this is a highly useful database with a
user-friendly interface for tracking national resource use and pollution indicators, using both production-
based and consumption-based perspectives, including their extent of 'decoupling' from economic activity
and UNDP's Human Development Index.

However, the SCP-HAT database does not include a global perspective or deprivation-based social
indicators, nor does it contain explicit ecological targets that could be used to assess decoupling in terms of
absolute sustainability, as far as we can see. For example, Germany achieves absolute decoupling of
consumption-based carbon footprint from economic activity over the 1990-2022 period (commendable!)
but is this decoupling sufficient to keep warming within its fair share of a safe climate boundary? Unless we
are mistaken, that question cannot be answered directly from the SCP-HAT database. Our point here is to
underscore that the scoping and selection criteria for a useful comparative analysis of the Doughnut to
'alternative' monitoring frameworks can be quite broad (with more possibilities for informative results, in
our view, in which case SCP-HAT could be included), or more narrow (in which case we believe the lack of
social indicators or explicit ecological targets in SCP-HAT could be grounds for exclusion).

All that being said, and with apologies for a lengthy response, we have revised the text by adding a brief
high-level discussion comparing the Doughnut to the SDG framework – the most similar alternative
framework, in our view – with reference to studies that conduct more in-depth comparisons of these two
frameworks (Lines 320-327). We believe this addition provides a useful complement to the similar
discussion on 'decent living standards' and 'safe and just Earth-system boundaries' that we added in the
previous round.

Regarding my comment about spillovers, I see now that you have included consumption-based indicators
(thank you). What I could not find though is any mention of spillovers. You deal with shortfalls and
overshoots, but some of those are strongly influenced by unequal trade relations, and outsourcings of the
North to the South, which is surely an important trend, contributing to why we're actually seeing the global
picture that you paint? I.e. social and environmental decline in the South driven by Northern affluence (see
here the Scientists's warning by Wiedmann). So I was surprised that such outsourcing does not get a
mention.

Thank you for this comment. We have revised the text to clarify that consumption-based footprint
indicators account for the outsourcing of upstream environmental burdens enabled by international trade
(Lines 98-100 and Lines 732-737)

Other than that - as I said before, I like your work!

Thank you very much for the time and care you have dedicated to reviewing our study.

**Referee 3**

The Doughnut framework is a compelling way to communicate information about sustainable development
that nicely includes both socioeconomic and ecological data. The revised version of the paper does a better
job of making the main points in a more compelling manner than the original draft.

We are grateful to the Referee for dedicating their time and expertise to our revised study.

The authors have done an adequate job of addressing most of my specific comments. The two exceptions
are on my original comments numbers (4), (6) and (7).

Thank you, we have made several revisions that we believe respond to these specific comments, as detailed
below.

In comment (4), I said that the planetary boundary framework is questioned, especially for indicators that
where the concerns are regional and not global in scale (N, P, HANPP). The authors did not address this
point but merely said they did not have space to do an "in-depth critical assessment." I think it is a problem
that deserves more attention when there are major flaws in an existing framework which the authors adopt
without thinking critically about how it might affect results. I would like to see the authors think more
deeply about their uncritical use of planetary boundaries. At this point, I do not insist that they revamp
their analysis, but that they at least recognize major issues and discuss this briefly in the paper.

We are grateful for this invitation to articulate our reasons for adopting the planetary boundaries
framework to define the Doughnut's ecological ceiling. We did indeed state that an in-depth critical
assessment was out of scope in our previous response to the Referee – and upon re-reading, we apologise
if our brief response appeared curt – but we also referenced the lines that we added to the revised
manuscript as a direct response: 'The dimensions of the Doughnut's ecological ceiling are defined by the
framework of nine planetary boundaries, and it is beyond our scope here to provide a critical assessment,
though we note that the framework has been the subject of considerable scientific scrutiny (see Ref.²⁶ for
an in-depth review).' In our view, although we may not have gone into as much detail as the Referee would
have liked to see, our response to this point in the previous round was more than what we perceive from
the Referee's characterisation.

Respectfully, we believe that variations of the regional concerns raised by the Referee were anticipated in
the original formulations of the framework by Rockstrom et al. back in 2009, especially the longer *Ecology*
& *Society* publication, and have been discussed at length in many publications since then (beginning with
the seven thoughtful commentaries published alongside the original *Nature* publication).

While we remain convinced that the scope of our study is not able to do justice to the depth and breadth of
the major issues in these scientific debates, we have revised the text to contextualise the huge uptake of
planetary boundaries and resulting scientific debate, while emphasising that our adoption of the planetary
boundaries to define the Doughnut's ecological ceiling is not taken lightly or uncritically – it is an intentional
choice based on our understanding of the state of the art in Earth-systems science, and crucially, our
aligned commitment to iteratively incorporate new knowledge as that science continually advances. We
also acknowledge that the limitations of the planetary boundaries framework could affect our results and
may require further investigation (Lines 600-609).

Point (6) is a fairly specific point, which I still don't understand. Why is malnutrition falling while food
insecurity is rising? I understand that these are different metrics and that this result is theoretically
possible, but still it seems improbably in reality and I would like to see a clearer explanation of what is going
on. Just saying it may happen doesn't really explain much for me.

Thank you for this invitation to reconsider our previous response to the seemingly paradoxical divergence
between the two food indicator trends. We have investigated in more detail to provide a clearer
explanation. In short, we believe these diverging trends result from two main factors: (i) the food insecurity
time series is considerably shorter than the undernourishment series (2015–2022 and 2000–2022,
respectively), which makes it more sensitive to the adverse impacts of the COVID-19 pandemic, and (ii) the
goal to eliminate undernourishment is a lower standard than the goal to eliminate moderate and severe
food insecurity. We have revised the relevant sub-section of Supplementary Information to elaborate these
points in more detail (Supplementary Information Lines 85-98).

On point (7), I don't think it clear that all environmental problems tie to over-consumption and affluence.
Specifically agricultural expansion, deforestation and other forms of habitat loss since 2000, which is the
primary driver of biodiversity loss, has primarily occurred in low-income tropical countries, not in high-
income temperate countries. Even middle-income countries, notably China, have transitioned to net
reforestation. I am not convinced that the authors revisions in the paper address my comments nor did
they do much to convince me or provide stronger evidence.

Thank you for this comment. We believe it is now well-established that increasing consumption/affluence,
including a growing middle class, is a primary driver of global ecological overshoot, which is our focus in this
study. However, we do not deny that other factors may be more important than consumption growth with
regards to specific local ecological concerns and contexts. We have revised the text to acknowledge that
our consumption-based approach to measuring ecological overshoot, based on country-cluster per capita
footprints with respect to downscaled planetary boundaries, should be seen as complementary and
additional to assessments of locally relevant pressures and thresholds. We also note that such local
concerns may be affected by more immediate factors, such as overexploitation, urban/agricultural
encroachment, pollution, and population growth (Lines 743-749).

In the broad sweep of the paper, however, these are relatively minor points and while I would like to see
the authors take a deeper look into these issues, I don't think this is enough to hold up the publication of a
paper that I think many readers will find highly useful and will probably be widely cited.

We appreciate the Reviewer's time and consideration throughout the review process, and we are grateful
for the encouraging remark about our study being useful to many readers (we hope so!).

**Referee 4**

The authors' responses and efforts to address several of the reviewer comments is greatly appreciated. At
this stage, I have three specific comments/suggestions that can be addressed without too much of heavy-
lifting.

1- The design of Figure 1 makes it look like the planetary boundaries for Ozone, air pollution, and Ocean
acidification entirely safe, which is a bit misleading. PB 3.0 published last year indicates that two of them
(air pollution, and Ocean acidification) are pretty close to being breached. Rockström, J., Donges, J.F.,
Fetzer, I., Martin, M.A., Wang-Erlandsson, L. and Richardson, K., 2024. Planetary Boundaries guide
humanity's future on Earth. Nature Reviews Earth & Environment, 5(11), pp.773-788.

Thank you for this comment. It is true that Figure 1 does not visualise the extent of 'undershoot' for
indicators that have not yet transgressed their respective boundaries. However, such information is readily
available in our study. Notably, this was a core design consideration that we aimed to make visible in Figure
2, which does clearly show the extent of undershoot in ocean acidification, air pollution, and ozone layer
depletion (along with numerical values provided in Table 2 and Supplementary Information/Data). In our

view, this complementary figure provides the visual information noted by the Referee, so we have not
altered the design of Figure 1.

2- The author's response on the use of Palma ratio is quite intriguing -- Palma ratio, which measures the
ratio of the share of GNI going top 10% to that going to bottom 40% income brackets, is an "economic"
measure while "social cohesion" is a sociological variable.

Thank you for this observation. Given our sense of the contested nature of disciplinary distinctions, we
have chosen not to name indicators as belonging to specific disciplines, and we have left the relevant text
as-is.

3- The move to present a disaggregated analysis of three subgroups is helpful. Additionally, the authors
should also acknowledge and flag for future research the critique that "the SJOS remains a very theoretical
concept because a match does not exist between the Planetary Boundaries and the social components of
the Doughnut." [Ferretto, A., Matthews, R., Brooker, R. and Smith, P., 2022. Planetary Boundaries and the
Doughnut frameworks: A review of their local operability. *Anthropocene*, 39, p.100347].

Thank you for this suggestion. We had attempted to communicate this point in our previous submission
although we did not flag it as an avenue for future research. We have revised Lines 338-342 to explicitly call
on researchers to address this important gap: 'A disadvantage is that there is no directly traceable link
between the social and ecological domains in the Doughnut, although frameworks are emerging that track
the global material requirements of achieving minimum standards for all of humanity⁴⁹, which we believe
are highly complementary and *important avenues for future research*.'

More broadly, we are very grateful for the time and care that the Reviewer has dedicated to our study.

**Referee 5**

I am unable to reproduce the first regression result, as detailed below.

Thank you for making time to attempt to reproduce our regression result for the undernourishment
indicator. Reproducibility concerns are very important to us, and we have investigated the Referee's claims
thoroughly, as described below.

A bigger problem however is the statement: "Values are reported as two-year averages." (p.11). If X_1 , X_2 ,
X_3 are independent random variables, then $\text{Cov}(X_1+X_2, X_2+X_3) = \text{VAR}(X_2) > 0$. Therefore the two-year
averages are serially dependent. The simple regression methods used here are not appropriate; one should
use proper time series methods. The effect of taking this correlation into account will be to reduce the
effective number of observations and hence increase the uncertainty in their estimated coefficients. A good
source, specially tailored to climate time series, is Leroy SS, Anderson JG, Ohring G (2008) Climate signal
detection times and constraints on climate benchmark accuracy requirements. *J Clim* 21:841–846.

Thank you for these observations. We regret the statement 'Values are reported as two-year averages' in
the table captions has apparently created confusion, as we were only referring to the numbers reported in
the '2000-01' and '2021-22' columns of Table 1 and Table 2. We have revised the captions to clarify. To
confirm, the time series of annual observations for each indicator was used to estimate the slope
coefficients reported in the 'Historical trend' columns (not two-year averages), so we have not introduced
serial dependence in the manner stated by the Referee.

That said, we acknowledge that such serial dependence may be an issue using annual data as well, so we
have revised our statistical analysis of historical trends to estimate and report standard errors that are
robust to heteroskedasticity and autocorrelation (Lines 669-671). We agree with the Referee that this is an
important feature to take into account, and we are grateful for the improved accuracy of our estimates.

Although these robust standard errors are generally larger than our previous estimates, we note that they
 are not large enough to alter the statistical significance of any of the indicator-specific trends reported in
 our previous submission (i.e. all of the time trends that were previously significant at 99% or 99.9% levels
 remain so).

Ignoring this, I attempted to reproduce the first regression in Table 1.

Dimension	Indicator	Shortfall (%)		Historical trend	To eliminate shortfall by 2030	
		(% of global population, unless otherwise stated)	2000-01	2021-22	(%pt per year)	(%pt per year)
Food	Population undernourished		13	10	-0.2** (improving)	-1.2

The slope or trend coefficient is -0.2.

"We estimated historical trends over time for each indicator of social shortfall and ecological overshoot
 using two-sided ordinary least squares regression with a linear model, or $y = \beta_0 + \beta_1 t$, where t is a year
 index with base year 2000, y is the indicator, and β_0 and β_1 are the regression coefficients (intercept and
 slope, respectively). See Supplementary Data for the full regression model results for each indicator,
 including estimated coefficients, standard errors, p-values, and coefficients of determination (adjusted r2)
 ". P(14). Neither I nor Google have heard of " two-sided ordinary least squares regression" though I have
 heard of two-sided hypothesis tests.

Thank you for this correction regarding two-sided hypothesis tests, which was an oversight on our part. We
 have revised this text and the table captions accordingly (Table 1, Table 2, and Lines 666-669, respectively).

The results are reported in 481643_1_extended_data_4469133_sfsg36.xls

481643_1_extended_data_4469133_sfsg36.xls					
	coefficient	SE	p-value	coefficient	adj-R2
(Intercept)	0.129	0.004848	4.69E-17		0.699
T	-0.003	3.95E-04	7.75E-07		0.699

Note that the trend coefficient is -0.003 and not -0.2.

It appears to be the case that the Referee used the previous Supplementary Data file from our original
 submission, not the most recent Supplementary Data file from our revised submission. On this point, we
 are very certain. We are not sure how this mix-up occurred, but the numerical values estimated by the
 Referee from the original submission are not comparable to the results reported in Table 1 of the revised
 manuscript because the underlying data have changed. Notably, we added an additional year extending the
 time series to 2022 in the revised version (previously 2021).

Using their data I ran the Linest regression package in XL and found

recomputed XLS linest					
	Coefficients	SE	p-value	coefficient	adj-R2
Intercept	12.8525692	0.484765	4.68607E-17		0.698748
t	-0.2785997	0.039515	7.74805E-07		0.698748

The P-value coefficient and adj R2 agree but the coefficients and SE's in the authors' table appear to be a
factor 100 too low. The units in their spreadsheet are clearly given as "percent", same as in Table 1. It
appears that they made a scaling error. The trend coefficient -0.2785997 is not the -0.2 mentioned in their
Table 1.

For reference, the trend coefficient the Referee estimated for population undernourished can be rounded
to -0.3, which is identical to the value we reported in Table 1 of our original submission. We have
reproduced the revised trend coefficient on the correct data file with no issues using their Excel-based
Linest methods (-0.243, or -0.2 with rounding).

With respect to the Referee's comments on our previous spreadsheet, there are four columns of relevance
to reference in response: 'value', 'boundary', 'unit', and 'ratio' (as mentioned in the 'Overview' tab). The
*units* that we report correspond to the *values* and *boundaries* reported for each indicator, not the
normalised *ratios* of overshoot/shortfall (which are dimensionless quantities/quotients without units
resulting from dividing *value* over *boundary*). We reported these normalised *ratios* in quotient terms in the
spreadsheet, partly to avoid confusion with the *units* reported for the *values* and *boundaries*, although we
see now that this was not clear.

In our previous submission, we conducted the regressions for each indicator using the shortfall/overshoot
ratios expressed in quotient terms (i.e. 0.13) rather than percentage terms (i.e. 13%). Although this may be
seen as a 'scaling error', in our experience it is quite common for either quotients or percentages to be
reported in databases (i.e. without or with multiplication by 100, respectively). However, we see now that
our previous approach could lead to confusion with the percentage-based results presented in the text
(including Table 1, and Extended Data Figure 5), and we are grateful to the Referee for identifying this
inconsistency in our reporting. We have revised our analysis to estimate historical trends and to express the
shortfall/overshoot values in percentage terms consistently throughout to avoid the risk of confusion.

I spent quite a bit of time trying trying to reproduce their results before concluding that they simply made
two compounding errors, a scaling error and a transcription error. I am not willing to review any
subsequent revisions of this ms.

We regret that the combination of the Referee using an incorrect data file from a previous round and our
reporting in quotient terms have led them to spend undue time attempting to reproduce our results. We
are confident that others will not experience these potential sources of confusion, and we are grateful for
the Referee's time.

**Referee 6**

The authors have addressed my comments mostly by discussing the limitations.

We are grateful to the Reviewer for their time and care dedicated to our Analysis. Respectfully, we do not
agree that our efforts to address the Reviewer's comments in the previous round were 'mostly by
discussing the limitations'. In our view, this perspective understates the Reviewer's contribution to the
revised manuscript. Notably, the Reviewer's first major concern in the previous round made the very good
and constructive point that direct comparisons of %pt trends between social and ecological domains should
be avoided, which we agreed and removed from the revised study (with a substantial impact on our
previous 'headline' results).

The comment of a better disaggregation, which was also made by other reviewers, was addressed by a new
analysis considering country income levels. However, this analysis was not about trends in time as the rest
of the analysis, but rather on the broad spatial patterns.

Thank you for this observation. The country-cluster analysis is for a single year because we have collected
comparable high-quality footprint estimates and per capita boundaries from a very recent study, also
published in *Nature*, which are only available for a single year. We have revised the text to explain to
readers that the compilation of comparable national time-series estimates of consumption-based
environmental footprints and per capita boundaries is a non-trivial exercise that was beyond the scope of
the present study (Lines 776-779).

As such, dividing countries into clusters (e.g., poorest 40%, middle 40%, richest 20%) and showing that
richer nations have smaller social shortfalls but larger environmental footprints, while poorer nations have
larger social deficits but smaller footprints, is not a new result. These findings echo a substantial body of
research in human development and sustainability science. I am not deep into this (more economic)
research, but e.g. references below come to mind.

Thank you for this frank assessment and for sharing the references below. We do not disagree that our
disaggregated results are consistent with earlier findings, but we wish to emphasise that our core aim of
the country-cluster analysis was to develop a ‘downscaling’ framework that could be coherently compared
to the global Doughnut in order to make inequalities visible that tend to be masked by global aggregates.
Such attention to cross-scale comparability in a concise and visual manner that is consistent with the
planetary boundaries framework and the Doughnut’s deprivation-based social foundation has not
previously been integrated in earlier studies, as far as we are aware. We have revised the text to clarify
these contributions (Lines 108-112).

Also, consumption-based assessments that include CO₂, water, nitrogen, and other resource footprints
have been used by multiple researchers to demonstrate how high-income nations “overshoot” their fair
shares of global environmental limits, whereas low-income nations remain within many planetary
boundaries but fail to meet basic social needs. This aligns with previous studies that downscale planetary
boundaries and compare them to observed consumption footprints at the national level.

We appreciate this observation, which seems similar to the previous point above. As far as we are aware,
previous national-scale studies have not explicitly made visible how available consumption-based
assessments relate to the nine planetary boundaries, including by showing the Earth-system processes that
are currently lacking data at this scale. Instead, studies tend to show only the relevant processes that they
are able to analyse, rendering invisible the domains that they do not analyse. So, while the results align
with previous studies, we believe that the visualisations in Fig 3 provide an important contribution by
making visible the current gaps in consumption-based data availability in a manner that is far more
consistent with the broader planetary boundaries framework.

Finally, the observation that wealthier groups—though much smaller in population—contribute a
disproportionately large share of resource use or pollution, while poorer groups account for the majority of
shortfalls in meeting social indicators, likewise reflects well-established patterns in the literature on
environmental justice and global inequality. Together, these points confirm that while the data might be
updated, and the specific breakdowns or indicators may differ, the broad conclusion is consistent with
earlier scholarship.

Thank you for this assessment. From our perspective, there are three points worth noting in response. First,
we are not aware of any previous studies that provide a dual comparison of indicator-specific shares of
total global social shortfall held by countries/groups alongside their respective contributions to total global
ecological overshoot. We are aware of several studies that calculate such proportional contributions of
total ecological overshoot metrics, but none that formally compare these to the similar but distinct concept
of proportional shares of total social shortfall. We have revised the text to clarify this novel contribution
(Lines 106-108).

Second, we believe the Referee's exclusive attention on our country-cluster analysis neglects additional
contributions that our study provides, notably the global scale analysis of historical trends across each
indicator of social shortfall and ecological overshoot over the 2000-2022 period, with accompanying
illustration of how these trends relate to the ambition needed for humanity to live within the Doughnut by
2050. This contribution could have been articulated more clearly in the previous round, in our view, so we
have revised the text to clarify (Lines 103-106).

Third, we are grateful for the Referee's comments regarding the novelty of our Analysis, which have led us
to improve the explanation of how our study contributes additional insights with respect to previous
studies in a new paragraph dedicated to this end (Lines 102-112). That said, we perceive an implicit stance
from the Referee's comments that presenting results which are not consistent with previous scholarship
seems to represent their standard for novelty. In our view, while disproving previous findings is certainly
one domain of scientific novelty, and although we accept that the standard for publication in this journal is
high, we believe that our study's consolidated conceptual framework, cross-scale comparable methods,
distinct metrics, updated data, and concise visual representation of new results constitute additional
domains of scientific novelty that are worthy of publication as an Analysis in this journal.

So, now I tend to agree with the more critical reviewers of the last round, that the study is a nice science
communication, but does not deliver substantially novel results of an original research article.

We appreciate this assessment. Apart from our responses to the Referee's previous points above, we note
that our study has been submitted as an Analysis, not an original research Article.

I haven't looked into the literature in depth, but it might be different if there are interesting trends also in
the disaggregated data by income levels (But an economist will be better able to comment on this.).
Actually, I do not understand why the authors do not report on trends at that level, while they do the global
level.

Thank you for this comment. We agree that exploring trends in disaggregated data is a worthwhile avenue
of future research. As mentioned in our response to the Referee's third point above, our core aim of the
country-cluster analysis was to make inequalities visible that tend to be masked by global aggregates by
developing a 'downscaling' framework that could be compared coherently to the global Doughnut
(including the ecological ceiling defined by the planetary boundaries framework). We believe that our
single-year country-cluster analysis accomplishes this aim concisely within very limited space constraints,
which represents a contribution compared to earlier work. As mentioned in our response to the Referee's
second point above, we have revised the text to clarify that compiling national time-series data on
environmental footprints and per capita footprints was beyond the scope of the present study (Lines 776-
779).

To be constructive, to my mind it could be a good idea to cast this paper into a Review paper in Nature
Reviews, certainly with a more comprehensive coverage of the past literature.

We are very grateful for the constructive comments and the Referee's time dedicated to reviewing our
study.

**References:**

O'Neill, D. W., Fanning, A. L., Lamb, W. F., & Steinberger, J. K. (2018). A good life for all within planetary
boundaries. *Nature Sustainability*, 1(2), 88–95.

Wiedmann, T., Lenzen, M., Keyßer, L. T., & Steinberger, J. K. (2020). Scientists' warning on affluence. *Nature
Communications*, 11(1), 3107.

Dorninger, C., Hornborg, A., Abson, D. J., von Wehrden, H., Schaffartzik, A., Giljum, S., ... & Fischer-Kowalski,
364 M. (2021). Global patterns of ecologically unequal exchange. *Ecological Economics*, 179, 106828.

Gidden, M. J., et al. (2019). Global emissions pathways under different socio-economic scenarios for use in
CMIP6. *Geoscientific Model Development*, 12(4), 1443–1475.

In addition, United Nations Development Programme (UNDP) Human Development Reports, consistently
show persistent deficits in health, education, and income in lower-income nations, in contrast to high-
income countries.

I am unable to reproduce the first regression result, as detailed below. A bigger problem however is the statement: "Values are reported as two-year averages." (p.11). If X_1, X_2, X_3 are independent random variables, then $Cov(X_1+X_2, X_2+X_3) = VAR(X_2) > 0$. Therefore the two-year averages are serially dependent. The simple regression methods used here are not appropriate; one should use proper time series methods. The effect of taking this correlation into account will be to reduce the effective number of observations and hence increase the uncertainty in their estimated coefficients. A good source, specially tailored to climate time series, is Leroy SS, Anderson JG, Ohring G (2008) Climate signal detection times and constraints on climate benchmark accuracy requirements, J Clim 21:841-846.

Ignoring this, I attempted to reproduce the first regression in Table 1.

Dimension	Indicator	Shortfall (%)		Historical trend	To eliminate shortfall by 2030
		2000-01	2021-22	(%pt per year)	(%pt per year)
	(% of global population, unless otherwise stated)				
Food	Population undernourished	13	10	-0.2** (improving)	-1.2

The slope or trend coefficient is -0.2.

"We estimated historical trends over time for each indicator of social shortfall and ecological overshoot using two-sided ordinary least squares regression with a linear model, or $y = \beta_0 + \beta_1 t$, where t is a year index with base year 2000, y is the indicator, and β_0 and β_1 are the regression coefficients (intercept and slope, respectively). See Supplementary Data for the full regression model results for each indicator, including estimated coefficients, standard errors, p-values, and coefficients of determination (adjusted r^2). P(14). Neither I nor Google have heard of "two-sided ordinary least squares regression" though I have heard of two-sided hypothesis tests.

The results are reported in 481643_1_extended_data_4469133_sfsg36.xls

481643_1_extended_data_4469133_sfsg36.xls					
	coefficient	SE	p-value	coefficient	adj-R2
(Intercept)	0.129	0.004848	4.69E-17		0.699
T	-0.003	3.95E-04	7.75E-07		0.699

Note that the trend coefficient is -0.003 and not -0.2.

Using their data I ran the Linest regression package in XL and found

recomputed XLS linest				
	Coefficients	SE	p-value_coefficient	adj-R2
Intercept	12.8525692	0.484765	4.68607E-17	0.698748
t	-0.2785997	0.039515	7.74805E-07	0.698748

The P-value coefficient and adj R² agree but the coefficients and SE's in the authors' table appear to be a factor 100 too low. The units in their spreadsheet are clearly given as "percent", same as in Table 1. It appears that they made a scaling error. The trend coefficient -0.2785997 is not the -0.2 mentioned in their Table 1.

I spent quite a bit of time trying trying to reproduce their results before concluding that they simply made two compounding errors, a scaling error and a transcription error. I am not willing to review any subsequent revisions of this ms.